# The Molecular Biodiversity of Protein Targeting and Protein Transport Related to the Endoplasmic Reticulum

**DOI:** 10.3390/ijms23010143

**Published:** 2021-12-23

**Authors:** Andrea Tirincsi, Mark Sicking, Drazena Hadzibeganovic, Sarah Haßdenteufel, Sven Lang

**Affiliations:** 1Department of Medical Biochemistry and Molecular Biology, Saarland University, 66421 Homburg, Germany; s8antiri@stud.uni-saarland.de (A.T.); mark.sicking@uni-saarland.de (M.S.); drazenapervan@yahoo.com (D.H.); 2Department of Molecular Genetics, Weizmann Institute of Science, Rehovot 7610001, Israel

**Keywords:** endoplasmic reticulum, GET, protein targeting, protein transport, SND, SRP, Sec61 complex, EMC, positive-inside rule, hydrophobicity, signal peptide, transmembrane helix

## Abstract

Looking at the variety of the thousands of different polypeptides that have been focused on in the research on the endoplasmic reticulum from the last five decades taught us one humble lesson: no one size fits all. Cells use an impressive array of components to enable the safe transport of protein cargo from the cytosolic ribosomes to the endoplasmic reticulum. Safety during the transit is warranted by the interplay of cytosolic chaperones, membrane receptors, and protein translocases that together form functional networks and serve as protein targeting and translocation routes. While two targeting routes to the endoplasmic reticulum, SRP (signal recognition particle) and GET (guided entry of tail-anchored proteins), prefer targeting determinants at the N- and C-terminus of the cargo polypeptide, respectively, the recently discovered SND (SRP-independent) route seems to preferentially cater for cargos with non-generic targeting signals that are less hydrophobic or more distant from the termini. With an emphasis on targeting routes and protein translocases, we will discuss those functional networks that drive efficient protein topogenesis and shed light on their redundant and dynamic nature in health and disease.

## 1. Introduction

Eucaryotic cells use the principle of compartmentalization to streamline the flow of information within the crowded intracellular environment. Different subcellular compartments have occurred during evolution as the result of either endosymbiosis, invagination of the plasma membrane, budding off from other previously formed organelles, or, as discussed more recently, from the autogenous fusion of plasma membrane protrusions [1,2]. Irrespective of its inside-out (protrusions of the procaryotic plasma membrane) or outside-in (invagination of the procaryotic plasma membrane) origin, the lumen of the endoplasmic reticulum (ER) was at first similar to the extracellular milieu and therefore different from the cytosol. In mammalian cells, the ER lumen still reflects its extracellular derivation based on the high concentration and storage of calcium [3,4]. Next to the nucleus, the ER is one of the largest organelles in many cell types and is abundantly present in secretory cells such as those found in the endo- and exocrine portions of the pancreas [5,6]. The correlation between protein secretion and abundance of the ER is not coincidental and substantiates the importance of the ER for this process [7,8]. Indeed, the ER represents the entry point for proteins to the secretory pathway including the exocrine zymogens and endocrine hormones released by the pancreas.

About one-third of eucaryotic genes encode for polypeptides that require targeting to the ER membrane [9,10]. Those polypeptides belong to soluble, membrane-associated, or integral membrane proteins that are all handled and distributed by the ER. Although this organelle represents a vast network that spans from the outer nuclear membrane to the periphery, newly synthesized proteins do not find the ER autonomously. Instead, precursor polypeptides rely on specialized targeting mechanisms that direct them to the ER membrane in a co- or post-translational fashion [11]. In the case of co-translational protein targeting, a nascent polypeptide is recognized during the process of ribosomal translation. As soon as a specific amino acid stretch of the precursor, a so-called targeting signal, emerges from the ribosomal exit tunnel, it is recognized by a targeting factor that directs the complex of ribosome and nascent chain to the ER membrane. The signal recognition particle (SRP) was the first co-translationally acting targeting factor that was discovered and shown to target the ribosome-nascent chain complex (RNC) with the help of the cognate SRP receptor (SR) to the ER membrane [12,13]. In contrast, post-translational protein targeting occurs after a precursor is fully synthesized and released from the ribosome into the cytosol. Key features of post-translationally targeted polypeptides can be a short overall precursor length (<100 amino acids), a C-terminal positioning of a transmembrane helix (TMH) that serves as a targeting signal, or, as is the case for yeast, a “weak” N-terminal signal peptide (SP) that is required for targeting [14,15,16]. These features prevent efficient recognition by the co-translational targeting factor SRP and require the presence of a post-translational targeting factor such as GET3. GET is the acronym for guided entry of tail-anchored proteins. This implies that GET3 is involved in the targeting of tail-anchored (TA) proteins, substrates that carry a single C-terminally located TMH as a targeting signal. In addition, GET3 has been shown to promote the targeting of short precursor polypeptides with a cleavable N-terminal SP [17,18,19,20]. Recent evidence has uncovered another SRP-independent (SND) targeting pathway that seems to function as a backup system with some substrate spectra that overlap with the GET and SRP pathway [21,22,23]. Additionally, abundant cytosolic chaperones support the targeting of fully-synthesized precursor polypeptides, but they are not discussed here in detail [24,25].

After targeting to the ER membrane, co- and post-translationally arriving polypeptides rely on one of the protein translocation machineries residing in this membrane. Similar to the diversity of precursor polypeptides that are handled by multiple targeting pathways, multiple protein translocases have also evolved to support the insertion or translocation of proteins into or across the ER membrane [26,27]. As the ER membrane originated from the procaryotic plasma membrane, some of the translocation machines of the ER resemble their bacterial ancestors [28,29]. The first ER protein translocase that was described and studied in much detail was the Sec61 complex [30]. This translocase can open up a “fenestrated” conduit through the ER membrane for the translocation of SP-carrying proteins as well as the lateral release of TMHs [30,31]. Other precursor polypeptides, such as TA proteins, can make use of alternative translocation machines such as the GET1/2 complex or the ER membrane protein complex (EMC) [32,33]. Interestingly, precursor proteins relying on the GET1/2 complex or EMC require a differentially shaped opening compared to the membrane-spanning pore that is provided by the Sec61 complex. This shows that nature has apparently found more than one solution for polypeptides to traverse a membrane.

It is now half a century since Blobel and Sabatini published their first “tentative scheme” on what they would later coin the signal hypothesis. They speculated about the “role of the ribosome-membrane interaction in the vectorial discharge of proteins into the ER”, which “is indicated by (a) the close association of the nascent polypeptide chain with the ER membrane and (b) the close association of the large ribosomal subunit with this membrane” [34]. As no one size fits all, several types of targeting signals, targeting pathways, and protein translocases have since then been described, which demonstrate the diversity of transferring proteins from the cytosol to the secretory pathway. However, in light of the macromolecular crowding and potential off-target destinations, one underlying theme that unifies those functional targeting and translocation networks is “safety first”. During their journey from the inside of the ribosome to the ER membrane, the nascent polypeptides are handed over from one protected compartment and cavity to the next to avoid extensive folding and premature, unproductive interactions. After leaving the shielded environment provided by the ribosome, hydrophobic targeting signals are cradled by a binding pocket of one of the targeting factors before they are transferred to a narrow opening of one of the translocation machineries that acts as a membrane-integrated chaperone. We will summarize some of these protective micro-compartments and discuss key players of the functional targeting and translocation routes that chaperone polypeptides into or across the ER membrane.

## 2. Two Prominent Types of Hydrophobic Targeting Signals

According to the original concept of the signal hypothesis by Günter Blobel and David Sabatini, proteins themselves contain targeting information within their polypeptide sequence that directs them to the ER membrane. Fifty years ago in 1971, they wrote that a “nascent chain provides the structural conditions for its transfer into the membrane-bounded compartment of the ER”. That is, intrinsic, proteinaceous targeting signals govern the transport and localization of precursor proteins to the ER membrane where “the segregated chains may undergo the modifications (proteolytic cleavage, e.g., proinsulin; attachment of carbohydrate, e.g., immunoglobulins) required for secretion, storage, or disposal in the various intracellular membrane-bounded compartments” [34]. Upon arrival at the correct membrane, the targeting signals fulfill a second function. They can initiate the gating of a compatible protein translocase that temporarily opens a pore or somehow helps to overcome the energetically unfavorable process of protein insertion/translocation into or across a lipid bilayer. A wealth of data gathered over the past 50 years demonstrates the validity of this oligopeptide-based targeting concept which relies on intrinsic amino acid stretches of cargo proteins. In addition, we would like to refer the reader to compelling data and reviews highlighting the oligonucleotide-based targeting of mRNA, which instead relies on intrinsic nucleotide motifs. In the case of the latter, specialized membrane proteins recruit either mRNA to membrane-bound ribosomes or mRNA that is present in a RNC to the membrane [35,36,37,38,39,40,41]. 

Here, we will focus on two types of oligopeptide-based ER targeting signals, cleavable SPs and TMHs, that are encoded as part of the primary structure of cargo proteins and help to find the ER. Both variants of targeting signals share a hydrophobic core element that upon its emergence from the ribosomal exit tunnel is usually considered to initiate the process of selective targeting, translocation, and/or membrane insertion [42]. Other hydrophobic sequences, such as the C-terminal glycosylphosphatidylinositol (GPI)-attachment signal [43,44], may further shape the choice of a specific targeting pathway but do not act independently of an N-terminal SP. Moreover, adhering to the Latin prefix “trans”, we do not discuss the targeting of monotopic membrane proteins that do not fully traverse both halves of the lipid bilayer and instead insert via an intramembrane domain often showing a hairpin or amphipathic helix [45,46].

### 2.1. The Tripartite Nature of Cleavable Signal Peptides

Soon after describing their insights into the ribosome–membrane interaction in eucaryotic cells, Blobel and colleagues published a series of landmark papers that demonstrated the presence of N-terminal SPs to direct proteins to the ER via the cytosolic SRP, before they are cleaved off upon translocation [12,13,47,48,49,50,51,52,53,54,55]. Although it was originally assumed that many, if not all, SPs share the same consensus motif, analyses of a growing number of protein sequences over time showed the opposite: a remarkable sequence variation of SPs. However, upon closer inspection, some preserved features shared by all SPs became evident and pointed for example to the importance of the central hydrophobic h-domain, which is usually rich in leucine [56,57,58]. Leucine residues stabilize the formation of an α-helix, a feature that is relevant at different stages during protein biogenesis [59,60]. Studies addressing the physicochemical parameters of SPs found a typical overall length of about 20–30 residues and validated the h-domain as a hydrophobic α-helical stretch of ~7–15 residues, which is flanked by two shorter domains named according to their relative position the n- and c-domain. While the n-domain (~1–5 residues long) can carry some cationic residues, the c-domain (~3–7 residues long) is polar in nature and harbors a SP cleavage site recognized by a signal peptidase [61,62,63,64,65,66]. Thus, most SPs are tripartite in their design, which is themed “cationic-hydrophobic-polar” (Figure 1A). 

Looking at the ~3500 human SPs and their combined 85,000 amino acids that are annotated at the Uniprot database (www.uniprot.org, accessed 12 June 2020), the same conclusions about physicochemical properties arise. The average length of a human SP is 24 residues and more than 80% are between 15 and 30 residues in length (Figure 1B). The shortest SP of eight residues is found in the chymotrypsin-like elastase family member 1, CELA1. Considering the importance of hydrophobicity and the h-domain, it is not surprising that the most frequent amino acid found in human SPs is leucine with almost 25%. However, the frequency of individual amino acids varies depending on the position, congruent with the tripartite sectioning of SPs. For example, in position #2 right after the starting methionine, the most frequent amino acid is alanine (19%) followed by glycine, arginine, and lysine with ~10% each (Figure 1C). The frequent occurrence of arginine and lysine in this position reflects the observation of a positively charged n-domain. In position #12, which should report on the h-domain of many signal peptides, leucine is the most prevalent amino acid with 35%. As expected, the four charged amino acids, aspartate, glutamate, lysine, and arginine are less frequently found at this position and have a combined frequency of about 5% (Figure 1D). Photo-crosslink studies addressing the targeting and translocation efficiency of different SP variants have shown that the severe transport defects of reporter proteins mainly derive from the shortening of the h-domain [58]. These findings underline previous studies that have tested the functionality of essentially random, 20 amino acid long peptide sequences for secretion [67]. Similarly, the occurrence of charged residues in the h-domain of a preprotein’s SP can affect translocation and cause disease, as is the case with mutations identified in the SP of renin or bilirubin UDP-glucuronosyltransferase [68,69,70,71,72]. Equivalent to charged residues, the introduction of a single helix breaking proline in the middle of the h-domain can also suppress the efficient translocation of the preprotein, as was shown for the SP of the yeast invertase [73]. The lack of a universal consensus motif and the detrimental impact on protein targeting in the presence of single mutations in the SP support the conclusion that SPs and the respective downstream mature domain of a protein are specifically matched. Hence, SP diversity does not represent a lack of selective evolutionary pressure. In fact, the opposite is the case and SPs have been found to evolve half as fast as neutral sequences which probably helps to conserve SP functionality [74]. The co-evolution of a SP with the rest of the protein also supports data that show SPs and mature domains are not always readily interchangeable without impacting translocation efficiency [75,76,77]. Therefore, it is not surprising that only about 80 of the ~3500 human SPs are re-used once or more within the human proteome.

#### The Correlation between Signal Peptides and Downstream Processes

The broad biophysical similarities plus the individual SP-specific features characterize most SPs as unique cis-acting elements that shape the fate of a protein including its targeting, translocation, SP cleavage, and post-translocation events. These processes involve various trans-acting factors that directly or indirectly associate with the SP. In addition to cytosolic targeting factors and membrane-resident translocases that temporarily harbor the SP, further auxiliary components that fine-tune these steps have been identified over the past years [42]. As will be discussed in more detail in the following sections, the first elements that were shown to accommodate the h-domain of co-translationally targeted SPs were the transiently opening binding pockets of SRP and the Sec61 complex [78,79,80]. In yeast and mammalian cells, certain secretory proteins can also be directed to the ER membrane SRP-independently in a post-translational fashion. Likewise, the secretion of proteins to the periplasmic space of *E. coli* occurs prevalently post-translation. Interestingly, the hydrophobicity of post-translationally directed SPs seems to differ from those that are recognized co-translationally; however, there are species-specific differences. While in yeast post-translationally targeted SPs show a lower hydrophobicity than the co-translational counterparts, post-translationally transported small secretory proteins of human cells show the opposite: an above-average hydrophobicity [17,81]. Irrespective of the organism, the transport of post-translational substrates requires the Sec61 complex to associate with the auxiliary membrane proteins Sec62/Sec63 that act as allosteric effectors supporting the opening of the translocation pore [82,83,84,85,86,87,88]. Other than SPs with a hydrophobicity deviant from the norm, human SPs with an above-average portion of the helix breaking residues glycine and proline require the presence of another auxiliary component, the translocon-associated protein (TRAP) complex [89]. Overall, the plethora of different SPs require the presence of multiple targeting and translocation modalities and provide one reason why the original idea of a unified pathway for the topogenesis of all secretory proteins had to be extended.

### 2.2. Transmembrane Helices Are Efficient Targeting Signals

In addition to N-terminal SPs, hydrophobic TMHs can also serve as efficient targeting signals and are readily recognized and accommodated by different targeting routes and translocation machineries. The two types of targeting signals are not mutually exclusive. About 40% of the SP-carrying human proteins mentioned above also have one or more TMH(s). Vice versa, of the ~4800 human membrane proteins that are annotated at the Uniprot database, ~1350 proteins carry a SP and ~50 a so-called transit peptide that is destined for the targeting of a nuclear-encoded protein to an organelle other than the ER (www.uniprot.org/help/transit, accessed 14 October 2021). The number of annotated human membrane proteins aligns reasonably well with previous reports stating that membrane proteins correspond to 20–30% of genes encoded in a typical genome [90,91]. On the one hand, TMHs and SPs share (i) the functional equivalence as targeting and gating signals as well as (ii) the extensive sequence variability. On the other hand, they differ with respect to (i) the length of their hydrophobic core and (ii) their positioning within the precursor protein. While SPs are found at the N-terminus and their h-domain comprises ~7–15 residues, TMHs can be localized anywhere within the primary structure and are typically 19–25 residues in size. Although TMHs are on average more hydrophobic than SPs, both types of targeting signals cover a wide range of hydrophobicity, a parameter that equally influences targeting and topogenesis [92,93].

A crude inspection of the ~4800 human membrane proteins annotated at the Uniprot database (accessed 19 May 2020) shows roughly 2100 (44%) bitopic/single-pass and 2700 (56%) polytopic membrane proteins with a totality of ~20,000 TMH (Figure 2A). A total of 88% of all human membrane proteins contain less than 8 TMHs and the shortest one with a total of 31 amino acids and 1 TMH is sarcolipin, a regulator of the ER calcium ATPase SERCA2. Without much deviation, the average length of all first, second, or third TMHs is 21 residues, a number that remains fairly stable even for membrane proteins with many more TMHs (Figure 2B).

#### 2.2.1. The Topological Diversity of Transmembrane Helices

The topological layout is one of many ways to classify membrane proteins. With a focus on TMHs, the following classification excludes β-barrel membrane proteins that are not present in the ER and are limited to the outer membranes of mitochondria, chloroplasts, and bacteria [94,95]. The topology-based grouping (Figure 2C), which is used by many researchers, considers four features including the presence of an SP, the orientation of the first TMH fully traversing the lipid bilayer, the relative position of this first TMH within the primary structure, and the quantity of TMHs giving rise to seven classes of α-helical membrane proteins [96]. Accordingly, single-pass type I membrane proteins start with a cleavable SP dictating the downstream TMH into an N_exo_/C_cyto_ orientation, whereas the type II version lacks a SP and the single TMH adopts an N_cyto_/C_exo_ orientation requiring an originally unanticipated flip-turn during insertion [97,98]. Interestingly, as targeting equivalents, the SPs and type II TMHs adopt the same N_cyto_/C_exo_ topology using a loop-like insertion or flip-turn. As will be discussed later, this is the reason why SP-carrying substrates (including soluble and type I membrane proteins) and type II membrane proteins share the same translocation apparatus, the Sec61 complex. In contrast, single-pass type III membrane proteins lack an SP and the single TMH orients in an N_exo_/C_cyto_ topology, which, whilst somewhat counter-intuitive, relies preferentially on a different translocation apparatus, the EMC [99,100]. Single pass type IV membrane proteins are more often referred to as TA proteins. They resemble the N_cyto_/C_exo_ orientation of a type II protein, but their TMH is located close to the protein’s C-terminus. This classification system for single-pass membrane proteins is also used at Uniprot (www.uniprot.org/locations/SL-9905 for type I and up to/SL-9908 for type IV). Based on the presence of an SP and the orientation of their first TMH, polytopic membrane proteins can also be classified analogously as type I_pt_, type II_pt_, and type III_pt_. We use the index “pt” to indicate the classification of a first TMH of a “polytopic” membrane protein.

#### 2.2.2. The Crosstalk between Transmembrane Helices and Trans-Acting Factors

The determination of the final topology of a membrane protein is a multifactorial process and depends on both trans-acting factors such as protein translocases, chaperones, and lipids as well as intrinsic parameters encoded within the protein sequence (Figure 2C). The sequence-intrinsic, or cis-acting, elements that influence the orientation of the initial TMH include the positive-inside rule, the folding capacity of preceding N-terminal domains, hydrophobicity, and the length of a protein reflecting on the duration of ribosomal translation [95,101]. The positive-inside rule is a driving factor for the orientation of both targeting signals, TMHs (initial and internal ones) and SPs. This conserved principle reflects on the statistical enrichment of arginine and lysine residues in the cytosolic edge of a TMH as well as in the cytosolic-facing n-domain of SPs [63,93,102,103]. Mutational and protein engineering analyses of membrane protein reporters have provided additional support for the bioinformatic approaches and corroborated the importance of positive charges as a topological determinant [104,105]. Larger, quick folding N-terminal domains that precede a TMH can start to adopt secondary and tertiary structure motifs that prevent their translocation and handling by the limited pore size of a translocase. Therefore, folded domains force the downstream TMH to adopt a type II (N_cyto_/C_exo_) topology, irrespective of the positive-inside rule [106]. The hydrophobicity of TMHs varies to a great extent. The TMHs of single-spanning membrane proteins are usually more hydrophobic than those of polytopic membrane proteins [107,108]. Similar to a quick folding N-terminal domain, a very hydrophobic TMH can also overpower the positive-inside rule. However, in this case, the hydrophobicity forces the TMH into a type III (N_exo_/C_cyto_) orientation by stimulating the translocation of the N-terminus [98]. Although the experiments by Goder and Spiess were based on an unusually hydrophobic 22 residue long oligo-leucine stretch, they impressively demonstrated two points. One, the total protein length following the TMH influences the topology, most likely as a function of synthesis time/speed. Two, the topology determination is a dynamic equilibrium that can entail TMH inversion starting from a type III insertion and culminating in a type II re-orientation. Furthermore, slowing down translation by low-dose cycloheximide treatments to increase synthesis time also favored TMH inversion from type III to type II. Those data aligned well with experiments using reconstituted conditions that also demonstrated the flip-turn of a canonical type II TMH from aquaporin IV [97,98]. 

In the case of polytopic membrane proteins, the topology determination of the first TMH dictates, in most cases, the orientation of the following TMHs with each one crossing through the membrane in the alternate orientation as the predecessor. However, the mixture of more and less hydrophobic TMHs sometimes only separated by a short loop can put further steric constraints on the topology determination and additional parameters such as TMH–TMH interactions come into play [109,110]. The cooperative behavior of neighboring TMHs can be deduced from experiments with wild-type and mutant proteins. For example, studies of the bovine heptahelical opsin show that, within the same polypeptide chain, some TMHs tend to insert into the lipid bilayer individually, whereas others bundle up before being released as a small package of TMHs [111,112]. Other studies that have focused on charge inversion mutations in the TMH-flanking regions of Glut1 or single point mutations in TMHs of connexin Cx32 have also demonstrated the cooperativity between TMHs [113,114,115]. Those findings imply that downstream TMHs can re-initialize correct topology determination and mutations do not necessarily cause an inverted topology of downstream TMHs or of the entire polytopic membrane protein. Less hydrophobic TMHs, sometimes also carrying membrane-aversive charged amino acids, can be found in ion transporters or channels such as the ER calcium ATPase SERCA2 or the voltage-gated potassium channel KCNA1 of humans. Such specialized membrane proteins have hydrophilic residues of different TMHs facing inwards, i.e., away from the hydrocarbon core of the bilayer. In the case of the potassium channel, TMHs and the hydrophilic residues form a tunnel with a selectivity filter and a gated pore as a conduit for potassium ions after shedding the water-shell [116]. With regard to cooperativity, the membrane-insertion of such “unusual” TMHs might require their release into the membrane in small bundles and the assistance of different trans-acting factors. 

Although instances are known where TMHs such as the type IV tail-anchor of Cytb5 can insert into protein-free membranes of liposomes [117,118], it is safe to say that other than the local lipid composition, protein translocation machineries also represent relevant trans-acting factors that help to shape topogenesis. As such, they decode the sequence-intrinsic information that influences the orientation of the initial TMH [119]. By analogy, protein translocases act in a similar way to the axon hillock of neuronal cells that collects, considers, and computes incoming inhibitory and activating signals from the dendrites into one unified signal output for the axon. Yet, before reaching a protein translocase at the ER membrane, the crowded intracellular environment requires the hydrophobic targeting signals to be cradled from the start to the end of their journey to prevent aggregation and premature, unproductive interactions. After the emergence of a targeting signal from the protective ribosomal exit tunnel, it is chaperoned by an adequate targeting pathway that safely transfers it to the membrane and a suitable protein translocase. Critical elements of the multiple targeting and translocation pathways that are required to properly accommodate the various targeting signals will be discussed next.

## 3. Multifactorial Polypeptide Targeting Pathways: The SRP, GET, and SND Systems

For years, the view of ER targeting was dominated by one pathway that would cater to all the polypeptides with the help of the SRP (Figure 3). The discovery of the GET and SND routes in the last fifteen years has led to the emergence of alternative targeting pathways [18,19,21,22]. Their existence has long been anticipated since the disturbance or absence of the SRP pathway does not result in a complete shutdown of ER targeting [120,121]. Differential energy requirements have been observed in the transport of different cargo [122,123,124] and the vitality of SRP knock-out cells also argued against its exclusive role in ER targeting [121,125,126]. However, the identification of the GET system as a dedicated pathway for TA proteins was still not sufficient to explain ER targeting of all cargo, hinting at the existence of additional pathways. Accordingly, the knock-out of GET components tends to have rather moderate effects, though heterogeneous phenotypes have been observed for various factors and organisms [127,128,129,130]. In line with the notion that GET is not essential in yeast [19], another targeting option was found with SND. Synergistic effects have been observed upon the simultaneous depletion of GET and SND and strikingly, an overexpression of SND could compensate for the loss of SRP function in yeast [21]. The proposed model involves a network of three main targeting pathways that show different, but partially overlapping, client spectra which are in accordance with a non-hierarchical organization where one pathway acts as a back-up in case of the malfunction of another. Specifically, SRP recognizes cleavable SPs and TMHs at the N-terminus, whereas the GET pathway preferentially binds the C-terminal TMHs of TA proteins. While both groups of clients share signals with a similar hydrophobicity but a different localization within the polypeptide, the SND pathway catches less hydrophobic targeting signals which occur internally or at the C-terminus [22,23,43]. These findings have highlighted a previously unanticipated complexity of ER targeting that expands beyond proteins with classical targeting signals and includes the heterogeneous cohorts of TA and GPI-anchored proteins [23,43,131]. Next, the current view of pathway organization and operation will be discussed for SRP, GET, and SND before we continue with a structural comparison of the central cargo binding components to discuss the partially overlapping client spectra.

### 3.1. The Ancient Silk Road Project, SRP—Direct Way to the ER

In the SRP pathway (Figure 3), two GTPases, the cargo-binding SRP in the cytosol and the corresponding ER-resident receptor SR, ensure efficient and specific targeting to the ER membrane [135]. Energy consumption is coupled to the dissociation of the SRP–SR complex upon the coordinated transfer of cargo to the Sec61 complex [136,137,138]. SRP is thus recycled into the cytosol for another round of ER targeting only upon the successful termination of the cargo delivery. Underlining its relevance as an important factor in ER targeting, the SRP pathway is conserved in all domains of life [139]. Though, similar to the ribosome, the size and complexity of SRP architecture has increased during evolution [140]. In contrast to the bacterial SRP that only comprises one protein component for cargo recognition (an SRP54 homolog), the mammalian SRP consists of six proteins bound to a universal 7S RNA scaffold. Along with the growth in size and complexity, eucaryotic SRP has developed the ability to arrest translation via the acquired Alu domain (SRP9/SRP14) that binds near the peptidyl transferase center (PTC) and competes with elongation factors [141,142]. It is based on this dogma that ER transport has long been assumed to be strictly co-translational, which entails the coupling of protein synthesis and translocation for maximal shielding of the hydrophobic targeting signal until translocation is completed. In an alternative model, the SRP-mediated slowdown of translation serves to keep the translocation competence of polypeptides to match the limiting targeting sites in the ER membrane, referring to the SR [143]. Today, the view of a strict translational arrest in SRP-mediated targeting has been weakened based on ribosomal profiling experiments in yeast and bacteria [125,144]. Indeed, SRP might not be the only option to locally slow down the translation, as mentioned before. The complete ribosome-independent action of SRP has likewise been proposed based on the binding to certain TA proteins [145], which seemed to hold functional relevance under cellular conditions [23]. In addition, the question of where the ribosomal translation is localized, in the cytosol or at the ER membrane, is an ongoing matter of debate. Accordingly, SRP might be prevalently required for the delivery of the RNC in the first round of ER targeting which is accompanied by local translation initialization at the ER membrane [146,147,148,149]. Breaking with another dogma, SRP seems not to be the lonely player it was generally believed to be. Mammalian small glutamine-rich tetratricopeptide repeat co-chaperone alpha (SGTA) has been shown to act co-translationally and downstream of SRP to ensure targeting fidelity [150], yet an overlap between the SRP and GET pathways has been demonstrated.

### 3.2. The Juggling GET Cascade—Safety First

The GET pathway (Figure 3) is well-established to deal post-translationally with the C-terminal targeting signals of TA proteins or short SP-carrying polypeptides with 100 amino acids or less, both of which are likely to evade co-translational recognition [22]. Despite some compensating mechanisms (see Section 4.1), TA proteins and short secretory proteins are released from the ribosome shortly after their synthesis is terminated without prior exposition of the targeting signal to the cytosol [151]. In such cases, a cascade of trans-acting factors handles the recognition, shielding, and delivery of the cargo [152,153]. Directionality of targeting towards the ER membrane is ensured by the homodimeric ATPase GET3 (Get3 in yeast), the central energy-consuming targeting factor of the GET pathway. Depending on the nucleotide occupancy, it transitions between closed and open conformations that allow cargo binding and release [154,155]. ATP hydrolysis is induced upon capture of the cargo and may represent a proof-reading step leading to full commitment to GET-driven ER targeting [155,156]. Thus, cargo delivery and factor recycling are carefully regulated to prevent premature cargo release before reaching the ER receptor. While such energy-driven control appears as a general theme in protein targeting, recent findings have shown that GET3 is dispensable for the targeting of moderate or low hydrophobic TA proteins. Since they are less at risk for aggregation, a loose binding and re-binding by cytosolic chaperones such as the abundant calmodulin might be sufficient. Calmodulin directs TMHs of lower hydrophobicity to the EMC for insertion [157].

#### Installment of a Pre-Targeting Complex with Connections to Other Chaperone Systems

In the canonical GET pathway, SGTA (Sgt2 in yeast) transfers cargo to the GET3 targeting factor as part of the so-called pre-targeting complex, together with GET4 and GET5 (Get4 and Get5 in yeast) [158,159]. While GET5 binds SGTA, GET4 recruits ATP-bound GET3 and primes it for capturing cargo from SGTA by the inhibition of ATP hydrolysis. Thus, by regulating the ATPase cycle of GET3, the pre-targeting complex supports the safe and direct loading of cargo from SGTA onto GET3. Such a hand-over is therefore characterized as private, in contrast to a simple release and diffusion of the cargo. Regulation of the GET3 conformation and cargo capture is hence driven by the nucleotide status and the interactions with effectors and cargo. In mammalian cells, the additional protein BAG6 (BCL2-associated athanogene 6) connects GET4 and GET5 to form the pre-targeting BAG6 complex (Figure 3). The C-terminus of BAG6 is involved in the scaffolding and has been shown to be the minimally required domain for TA targeting [160]. Its other domains recruit the E3 ubiquitin ligase RNF126, heat shock proteins (HSPs), and proteasomal subunits or bind misfolded proteins. Thus, BAG6 links the GET pathway to protein degradation [161,162,163,164,165]. Strikingly, BAG6 is antagonized by SGTA, which may involve the cycling of hydrophobic domains between both chaperones until the cargo is either triaged to the ER (via GET3) or the proteasome (Figure 3) [166,167]. The competitive binding between the various factors in building a complex that either drives targeting, or degradation is decisive for its fate. However, SGTA shows the fastest binding kinetic to the cargo, making it a central player of the GET cascade [166,167,168,169]. Thus, BAG6 may only have the chance to capture the cargo from SGTA when transfer to GET3 is failing and the cargo undergoes unproductive cycles of release and re-capture by SGTA. Engagement of the cargo is further competed for by calmodulin, an off-pathway chaperone preventing the degradation of the cargo and providing a safeguard of the GET pathway [161].

Likewise, SGTA-mediated sorting may involve the targeting of TA proteins to mitochondria, since the abolished delivery to the ER results in mistargeting to the mitochondria congruent with a default destination [19,170]. Indeed, BAG6 and SGTA are platforms for the recruitment of other off-pathway chaperones, such as Hsp70 and Hsp90, which have been associated with mitochondrial TA protein biogenesis [171]. However, the nature of the mitochondrial targeting pathway is still unclear and alternative implications for the connection of the GET pathway with heat shock proteins have been discussed. A recent yeast model suggested that soluble TA proteins may first be captured by Hsp70 for hand-off to Sgt2 and the subsequent transfer to Get3 [172,173]. Such funneling through a chaperone-guided cascade has been proven to be more efficient and protective against aggregation than the immediate loading of cargo to Sgt2. How this model applies to the mammalian GET pathway remains to be clarified, since whether SGTA and BAG6 directly bind the ribosome with putative implications for initial cargo binding has been discussed (see Section 4.3). Nevertheless, the Hsp70-Sgt2-Get3 triad provides an attractive model for how directionality is maintained in the chaperone cascade [152]). Hsp70 is highly abundant in the cytosol and binds rapidly but loosely to its cargo which subsequently engages stronger interactions with chaperones downstream in the pathway. The initially high binding kinetic might be required to compete with unfavorable misfolding and aggregation, while unidirectionality is later driven by thermodynamic forces ensuring high affinity and stable binding that prevent the reverse transfer to upstream components. This selective hand-over of cargo polypeptides in the targeting cascade from one chaperone to the next reminds us of the directed flow of electrons in the electron transfer chain from the lower to the higher redox potential.

The many factors acting in concert in the GET pathway raise the question of a rationale for such complexity, whereas SRP can fulfil ER targeting in a one-factor show. The multitude of components might support the rather transient and low-affinity interactions in the GET cascade [174]. The characteristic branching into diverse off-road pathways, however, demonstrates the importance of quality control in post-translational targeting. To compensate for misfolding and mistargeting, quality control systems are installed on the road and at the destination [161,175]. The modular organization further provides flexibility and options for internal rearrangement to bypass overloaded components or to allow a targeted regulation of defined cargo pools. On the other hand, SRP and GET might be representations of different targeting concepts that have co-evolved according to the needs of their cargos. They either provide one-to-one guidance for highly hydrophobic cargo in a direct delivery process (SRP) or tolerate the fact that the outcome is a balance between delivery and degradation of the cargo (GET), which may include shuffling between chaperons and moratorium stages in case of malfunctions or stress and starvation [176]. Metazoan GET pathway components are additionally connected to other cellular processes (see Section 5.2 and Section 6). GET2, the mammalian subunit of the GET receptor, is described as a signaling molecule and is associated with cell survival in the immune system, independently from TA biogenesis [177,178,179]. These GET independent functions might explain the controversially discussed four- to sevenfold molar excess of GET2 over GET1, its partner in the receptor complex [133,134,180]. GET5 is also related to other cellular functions and dysfunctions including actin cytoskeletal reorganization, cell migration [181], tumorigenesis [182], and bone development [183]. Those effects occur independently of its action in the GET chaperone cascade. 

### 3.3. The Lonely SND Player—Mysterious All-Rounder

Congruent with the observation of protein transport in the absence of SRP and GET [120,121], a genetic screen in yeast uncovered the SND pathway as an alternative targeting route capable of delivering a broad range of substrates to the ER. So far, three proteins called Snd1, Snd2, and Snd3 have been assigned as components of the SND route [21]. Initial evidence for the involvement of the *SND2* and *SND3* genes in protein targeting arose from another yeast screen showing synthetic lethality when deletions of either component were combined with deletions of the *GET2* or *GET3* gene [184]. Further, SND overexpression has been shown to have the capacity to rescue a SRP growth defect in yeast. This led to the working model of alternative targeting pathways which have overlapping client spectra and compensate for each other. Strikingly, SND as a backup route to the ER seems to be conserved in human cells [21,22]. While a detailed mechanistic description of the pathway is still missing, the currently known client spectrum in mammalian cells comprises co-translational cargo such as membrane proteins with internal TMHs and GPI-anchored precursor proteins with a GPI attachment signal of low hydrophobicity [22,43,185]. TA proteins have likewise been found among its clients when the GET pathway was compromised [22,23]. The three components attributed to the yeast SND pathway include the ribosome-binding factor Snd1 and the heterodimeric ER-resident receptor Snd2/Snd3 [21,186]. TMEM208 was identified as the human Snd2 homolog hSnd2 (Figure 3) [22]. Two sequence paralogs of Snd2 have also been identified in plants, which await functional validation [187]. In contrast to the GET cascade, no cytosolic factor has so far been characterized in mammalian cells either for initial signal recognition or for the chaperoning or directed delivery to the ER membrane. Assuming a similar case as for GET2, where a missing sequence similarity with Get2 complicated its identification, database searches for structural homologues might be instrumental in finding the missing SND components in higher eucaryotes [188]. Eventually, a targeting factor of another pathway, such as SRP, might support SND-mediated targeting at least in some cases. The recent observation of a dual recognition of GPI-anchored proteins at both termini may argue for such a scenario [43], where the C-terminus is captured by an SND component and the N-terminus by SRP for subsequent targeting to the Sec61 complex. Such duality might demonstrate another example of a mixed targeting reaction where modules from different pathways are shuffled together to fulfill a joint targeting activity. In comparison to the model of alternative targeting pathways backing-up each other, single components may also back-up shortcomings in other pathways. Further, Talbot et al. have shown a connection between hSnd2 and the EMC complex, which comes with several implications [185]. First, the SND pathway might target cargo to different ER translocation machineries such as the Sec61 complex or EMC (see Section 5.3). Second, combined with the observation that calmodulin and SGTA act upstream of EMC, the SND receptor component may team up with different targeting factors and translocases depending on the type of cargo. Another explanation for the lack of a cytosolic SND component in higher eucaryotes implies that hSnd2 may initiate a localized translation at the ER membrane which would bypass the challenges of cargo shielding and targeted polypeptide delivery. Lastly, a combination of the above scenarios is possible, assuming multiple roles for hSnd2 or the corresponding SND receptor.

### 3.4. Comparison of Signal Recognition by the Various Chaperones Involved in Targeting

Looking at the diversity of ER targeting signals (see Section 2), hydrophobicity appears as a central feature. Accordingly, trans-acting factors that mediate targeting to the ER membrane show hydrophobic pockets or binding grooves adapted to the properties of their clients for selective and efficient recognition. In the following, we shortly discuss structural analogies between the various components for cargo binding as well as mechanistic similarities or differences in targeting. Although information about the initial capture of cargo is missing for the SND pathway, understanding the limitations of SRP and GET sheds light on the putative capacity of SND as a back-up system.

#### 3.4.1. Cargo Capture by SRP

SRP binds highly hydrophobic cargo with the methionine-rich M domain of its SRP54 subunit [189]. Its hydrophobic groove is built by five helices and a connecting finger loop which supposedly provides high flexibility for the accommodation of various SPs and TMHs [78,190]. Cryo-EM structures of SRP engaged with a 21-residue TMH have revealed a density in the hydrophobic groove that has been attributed to an α-helix of ∼12 residues, corresponding to half of the TMH [191]. Assuming a similar mode of binding for cleavable SPs, the same length would account for the central h-domain as a crucial feature in cargo recognition. The M domain sits on the SRP RNA and is structurally connected via a flexible linker to the GTPase domain of SRP54 with implications for the further regulation of cargo binding [78,192]. Indeed, client specificity is provided by electrostatic interactions of the SRP RNA with charged residues in the targeting signal, such as the N-terminal polar region of cleavable SPs. Communication with the adjacent GTPase domain helps to coordinate between cargo and receptor binding. Mechanistic details revealed by Cryo-EM shed light on the hydrophobic groove which is pre-formed in the absence of a targeting signal and is shielded by amphipathic helices [191]. The binding of a hydrophobic signal displaces amphipathic helix 2 which then covers the targeting signal in a similar way to a protective lid. Intriguingly, this placeholder helix sets at the same time a hydrophobicity threshold for regulating the admission of cargo to the hydrophobic binding groove of SRP. 

#### 3.4.2. Cargo Capture by GET3

In the case of GET3, a composite hydrophobic groove is built during dimerization involving extensive interactions between the ATPase domains [193]. Such architecture allows communication between the nucleotide and TMH binding sites for the coordinated capture and release of cargo and the safe delivery from the upstream pre-targeting complex to the downstream ER receptor [33,194]. For instance, ATP binding stabilizes the assembly of hydrophobic helices at the dimer interface required for cargo binding. These helices are rich in methionine residues and those essential for TA binding have been attributed to the so-called “GET3-insert” including helix 8. It shows moderate evolutionary conservation with a varying length and methionine content. In animals, helix 8 has 3 methionine residues among 21 amino acids, while there are only 2 methionine residues among the 15 amino acids of the homologous helix in *S. cerevisiae* [195]. The apparent acquisition of additional methionine residues has been interpreted as a result of the specialization of GET3 in TA biogenesis which evolved from the bacterial ancestor ArsA, an arsenical pump without targeting activity. On the functional level, helix 8 has been described to act in a way similar to a dynamic lid that optionally shields the hydrophobic groove or the hydrophobic targeting signal from the aqueous environment. According to recent mechanistic insights, it engages the TMH first and guides it into the hydrophobic groove while preventing off-pathway chaperones from competing with Get3 for access to the TMH [196]. Strikingly, the lid accelerates the hand-over of cargo from Sgt2 to Get3 in cooperation with the pre-targeting complex. Thus, reminiscent of helix 2 in SRP54 setting the client specificity of the SRP pathway, helix 8 in GET3 may ensure privileged client transfer and pathway specificity in a network of chaperones. Interestingly, it has been debated whether the native targeting complex is comprised of a Get3 tetramer rather than a dimer, in which case a hydrophobic chamber is formed by two dimers allowing the binding of several signals and shielding all-round the TMH [194,197,198]. In this model, the role of helix 8 has been attributed to the stabilization of the tetramer interface. In contrast to these crystallography-based models where a hydrophobic chamber is formed for cargo binding in the closed state, a different model has been derived from single-molecule spectroscopy which has proposed the stable trapping of cargo in a dynamically open conformation of yeast Get3 acting as a “protean clamp” [199]. Here, the different modes of interaction allow the TMH to find the energetically best fit instead of locking into a fixed position in a pre-formed binding site. Moreover, helix 8 does not play a prominent role in this model. However, it solves the mechanistic dilemma of an exclusively closed Get3–cargo complex while allowing the high-affinity binding of the cargo with a dynamic and directed hand-over between factors. For the dimer, however, the overall hydrophobic area measures more than 3000 Å^2^ of surface which is twice the size of the binding groove in the M domain of bacterial SRP [200]. The ends of the hydrophobic groove are lined by charged residues, thus defining a length of 30 Å tuned to accommodate sequences of ~20 amino acids, such as the TMH of TA proteins destined for the ER [78,200]. Client hydrophobicity is comparable to that of SRP. Since the TMHs of mitochondrial TA proteins are typically shorter, less hydrophobic, have a lower helicity, and are C-terminally flanked by positively charged residues, GET3 may have some capacity to discriminate between TA signals of different organelles [201,202,203]. As shown by a systematic study in yeast, Get3 is indeed sensitive to TMHs with variations in hydrophobicity and helicity, thus representing one of multiple filters in the GET pathway for selecting cargo [156]. The physicochemical properties of TA-TMHs from ER, mitochondria, and peroxisomes overlap [153], yet the capacity to detect and select differential C-terminal charges seems to be missing in the GET pathway [156]. Yeast Get3 was indeed found to bind and deliver mitochondrial TA proteins to the ER, which led to the conclusion that an as yet undefined mitochondrial targeting pathway might outcompete the GET pathway under cellular conditions [204]. 

#### 3.4.3. Cargo Capture by SGTA

As the first selectivity filter in the GET pathway, SGTA has been shown to discriminate signals based on their hydrophobicity and helical content, including a certain distinction between ER and mitochondrial TA proteins [156,165]. Despite similar hydrophobicity thresholds of yeast Sgt2 and Get3, faster binding kinetics were assumed to increase the efficiency of Sgt2 in the rejection of unsuitable cargo [152,156,196]. Looking for features in yeast TMHs that allow better discrimination between different TA proteins, it was recently revealed that a so-called hydrophobic helical face geometry (the clustering of hydrophobic residues on one side of a TMH) was more successful in predicting the destined organelle than the overall hydrophobicity [205]. Strikingly, the finding was reflected by the results of a modeled structure and the binding behavior of yeast Sgt2. For human TA-TMHs, this correlation was less pronounced, hinting at a more complex scenario in the mammalian GET pathway. Here, a sequence of 11 residues within the TMH seems to be crucial for prediction, in combination with the C-terminal charge [206]. 

In contrast to the array of high-resolution crystal structures for GET3, the structural information of SGTA is limited. Homodimerization solely involves the N-terminus while the TPR domains in the center of the sequence typically interact with heat shock proteins [207]. Exploiting SAXS, NMR, and EPR experiments, the C-terminus has been characterized as functionally important in cargo binding containing one or two α-helical regions and a novel NNP and glutamine-rich domain [174]. In the proposed working model, the transient dimerization of the SGTA C-terminus results in the formation of a hydrophobic binding groove that captures cargo in a tweezer-like motion with all-around shielding of the TMH. How the binding and release of cargo is regulated without nucleotide binding and hydrolysis has yet to be determined. Recent computational modeling supported by biochemical data identified a similar hand structure for the binding of at least 11 residues, preferably containing clusters of leucine residues [205,208]. Since a typical TA-TMH has 18–20 amino acids, it was speculated that the two C-termini in a SGTA homodimer may occupy the full length of the signal, similar to the binding mode of calmodulin. However, a C-terminus is composed of five methionine-rich amphipathic helices forming a hydrophobic groove 15 Å in length. While a long N-terminal loop in yeast Sgt2 is supposed to shield the hydrophobic surface, the corresponding loop in human SGTA is shorter and may not be sufficient to complete the same action. Reminiscent of other chaperones in targeting, cargo binding leads to the stabilization of a rather unordered structure. By contrast, human SGTA appears to be more ordered with a higher content of glutamine residues, which may explain the differences in client binding, despite a similar hydrophobicity of the binding groove. Here, the binding to a more ordered structure causes lower entropic costs which may set a lower hydrophobicity threshold and, indeed, human SGTA was found to be less selective [205]. These data match with other findings indicating that SGTA contributes to the biogenesis of TA proteins with TMHs of high and low hydrophobicity [161]. How mitochondrial TA proteins are excluded from binding in the mammalian system remains to be defined in future studies. In sum, client specificity increases when moving downstream through the GET cascade. A similar working model is known from the HOP co-chaperone family that coordinates cargo transfer between HSP70 and HSP90 and to which an evolutionary connection has been suggested based on the similar domain organization of SGTA [208]. 

#### 3.4.4. Cargo Capture by BAG6

Signal recognition by BAG6 is widely uncharacterized due to missing structural information. Several domains at the N-terminus, two proline-rich and a novel BUILD domain, have been shown to build a platform for the binding of hydrophobic sequences in different contexts [209]. Poly-ubiquitinated proteins, defective proteins with exposure of short hydrophobic stretches, ERAD substrates, but also GPI-anchored and TA proteins are among its clients, with putative binding of the latter to a more central region of BAG6 [169]. How discrimination with this variety of clients is achieved is currently not known. For TA targeting, however, the BAG6 C-terminus which acts as scaffolding for the GET pre-targeting complex has been shown to be sufficient [160]. Although the capture of cargo by BAG6 seems not to be directly associated with targeting, rather than degradation, its competitive binding to GET clients might still contribute to their targeting fate.

#### 3.4.5. Cargo Capture by a Putative hSnd1

Sequence comparisons of the yeast counterpart have failed to identify the human hSnd1. Assuming that a functional homology may exist, the consideration of structural features could help to identify a cytosolic SND component in mammalian cells. The AlphaFold model of the 100 kDa yeast Snd1 protein shows a structurally complex molecule [210]. In the center of the folded molecule, a layer of seven β-sheets forms a platform that is partially confined by the charged residues of surrounding α-helices. Such a platform might be reminiscent of the hydrophobic groove described in other targeting factors. Yet, none of the 17 methionine residues of the Snd1 primary structure are part of the β-sheet assembly. Instead, the methionine residues appear in regions which are predicted with a low confidence score hinting at disordered domains that may stabilize and reconfigure upon cargo binding. Other characteristic structural motifs predicted for Snd1 comprise a 30 residue long coiled-coil domain in the middle of the sequence followed by an unusually acidic stretch. Structural studies of Snd1, eventually with a bound cargo, will elucidate the substrate binding mechanism and provide a rationale for the broad range of transported substrates.

#### 3.4.6. Common Cargo Capture Principles Used by Some Cytosolic Targeting Factors

Comparing the evolved binding sites in the different targeting components showcases the general mechanistic characteristics and the potential of the client spectrum of each pathway. For instance, weak arrangements of amphipathic helices rich in methionine residues that are stabilized upon client binding and the displacement of a regulatory lid helix appear as common themes. Differences become apparent when comparing the binding kinetics and the energetic aspects of the varying targeting factors, co-chaperones, and off-pathway chaperones. All these have effects on the functional outcome, directionality, and specificity of an interaction. For instance, SRP appears as dominating targeting factor with priority over GET, though both share binding grooves of similar hydrophobicity. Another example is given by SGTA, where the hydrophobicity of the binding site alone seems not to be decisive, rather than its degree of structural order. Since focusing on crystal structures of hydrophobic grooves is not sufficient to explain the client spectrum, novel kinetic approaches (single-particle spectroscopy, molecular modeling), systematic screening for clients (trap mutants), and advances in the bioinformatic determination of characteristic features in the targeting signal complement the current view of how cargo recognition defines the targeting fate. In the native cell, of course, the probability for a putative interaction is equally important in a network of competing chaperone systems. Thus, it is also decisive for the targeting choice which targeting factor is given primary access to the client. Helices of the binding pockets play a regulatory role in providing or preventing interactions. Chances for additional stimuli that can regulate the targeting fate are provided by the different energy requirements of the pathways (GTP versus ATP), with GET3 reversibly switching between a targeting and a holdase activity in accordance with the energy status of the cell [176]. *HSND2* expression is transcriptionally regulated in response to hypoxia [211]. Another option for regulation involves post-translational modification. The phosphorylation or acetylation of a targeting signal or the central NTPase domain could disrupt communication with the cargo binding site for the inhibition of client capture (cf. alkylation of SRP with NEM [189]) and would be reminiscent of the temporary phosphorylation of some transit peptides destined for chloroplasts [212]. Despite all control, mistargeting may occur, for which separate regulation systems have been established by the cell, such as for the removal of proteins from the mitochondrial outer membrane or ER membrane [213,214]. Moreover, the dual targeting, re-targeting, or SURFing of proteins from one organelle to another potentiate the diversity of targeting networks [153,215,216].

## 4. The Interplay of Targeting Factors at the Ribosomal Exit Tunnel

The biogenesis of proteins destined for entry into the secretory pathway starts at a cytoplasmic ribosome, a megadalton-sized ribonucleoprotein complex composed of core components plus associated factors (Figure 4). Their flexible recruitment reflects the functional connection of protein synthesis with protein folding, quality control, and downstream cellular processes [217]. Variability in the ribosomal composition has likewise been interpreted as a functional specialization of distinct ribosomal populations that selectively translate certain mRNA [218]. The impact of ribosome heterogeneity on localized translation (initiation in the cytosol versus at the membrane of specific ER domains) or protein targeting remains to be further investigated [36,146]. However, it is tempting to speculate that alternative mechanisms of translation initiation influence the choice of a protein targeting pathway. These involve elements in the mRNA, such as internal ribosome entry sites or RNA hypoxia response elements (rHREs), which may prevalently occur in cargos that utilize the same ER targeting route [219,220,221]. Endowing the corresponding targeting factors with the same mRNA control element (e.g., rHRE) would allow the coordinated regulation of both processes, translation and targeting (cf., oxygen-regulated transcription of *HSND2* [211]). Likewise, distinct elements in the mRNA may directly dictate the choice of a targeting pathway by the recruitment of a certain targeting factor. Such capacity has indeed been described for mRNA with certain 3′ untranslated regions and SRP [222]. Specialized ribosomes may thus be primed for a specific targeting pathway and the selective translation of specific mRNAs coding for respective clients. All these aspects are refining our current view of ribosomal protein synthesis towards a more active role of the ribosome in proteostasis.

### 4.1. Shaping the Polypeptide Fate from within the Ribosomal Tunnel

As the protein synthesis machinery, the ribosome represents the first interaction partner that a polypeptide encounters on its way to the ER. The growing amino acid chain is born at the PTC which is buried inside the ribosome and connected to the surface and the cytosol via the ribosomal exit tunnel. Its length is typically assumed to cover 30–40 amino acids of the nascent chain [223]. The important features are the two constriction zones inside the tunnel and a vestibule near the exit port (Figure 4). The latter is evolutionary less conserved than the beginning of the tunnel close to the PTC. Moreover, due to the presence of the eucaryote-specific eL39 protein, the exit region of the tunnel is more confined in eucarya than bacteria [224]. Considering the space limitations, polypeptide folding inside the tunnel has long been assumed to be restricted to the formation of α-helices [225,226]. In procaryotes, however, the folding capacity of the exit tunnel has recently been expanded to encompass tertiary structures and small protein domains [227,228,229]. Regulation of these events involves the interplay between the nascent chain and the geometry and electrostatics of the exit tunnel [230,231,232]. Specifically, the negatively charged tunnel interior affects the translational speed and thus, the elongation rate and folding kinetics of the nascent polypeptide chain. All of these may influence the downstream interactions of the RNC and the choice of a targeting pathway. First, a decreased translational speed expands the time window for the recognition of the cargo by cytosolic factors before the release of the polypeptide chain from the ribosome. By comparison, the rate of translation elongation is slow in eucaryotes where at the same time complex chaperone systems are associated with ribosomal protein synthesis. Consequently, it has been proposed that evolution has tuned translation elongation rates to coordinate chaperone binding [233]. Speed can also be regulated by ribosome-binding proteins, such as the SRP with its Alu domain acting on the PTC. Likewise, properties associated with the nascent chain itself can modulate translation kinetics. The local slowdown of translation by non-optimal codons strategically placed at a distance of 35–40 amino acids from the N-terminus has been shown to prolong exposure of a nascent targeting signal close to the tunnel exit and improve SRP-dependent targeting [234]. In another case, translational pausing during XBP1 synthesis supports membrane targeting to the Sec61 complex by the exposure of a targeting signal [235]. Polypeptide folding may also have the capacity to slow down translation from within the tunnel, as was observed for C-terminal TMHs of TA proteins. Such a mechanism compensates for the very limited time window of TA recognition upon release from the PTC and improves recognition by the targeting factor BAG6 [236]. Furthermore, emergence from the tunnel may be delayed in case of fast and excessive folding. Varying secondary structures show distinct escape kinetics with α-helices that are usually faster than β-sheets [237,238]. Moreover, folded states of the nascent chain inside the tunnel contribute to the selective recruitment of cytosolic targeting factors to the ribosome. In particular, TMHs of strong hydrophobicity have been found to stimulate SRP binding for the subsequent targeting of respective clients to the ER [191]. Hence, translational slowdown and the early recruitment of targeting factors mediated by the nascent chain or mRNA elements (see above) ensures proper recognition by downstream interaction partners, including factors for co- and post-translational protein targeting.

### 4.2. Allosteric Crosstalk from within the Tunnel to the Outside

Conserved positive charges near the ribosomal exit port attract various cytosolic factors that contribute to protein folding, targeting, and quality control. However, the ribosome tunnel exit and the ribosomal surface are not just passive platforms for diverse interactions with cytosolic proteins [239]. Some ribosomal proteins that line the tunnel interior reach the outside of the ribosome and shape its surface (cf. eL39, Figure 4). Such structural organization predestines the allosteric signaling from the RNC inside the tunnel to the cytosol, thus coupling, for example, ongoing protein synthesis with targeting by selected factors or post-translational modifiers [240,241]. Moreover, it was recently uncovered that a cytosolic chaperone that occupies the exit port, the nascent-polypeptide-associated complex (NAC, see Section 4.3), reaches deep into the tunnel to sense a nascent chain before it emerges into the cytosol [242]. Similarly, the bacterial SRP has been shown to penetrate into the exit tunnel and scan for nascent chains during ribosomal protein synthesis [243]. Hence, crosstalk from the RNC to the cytosolic targeting factors may involve ribosomal tunnel proteins including NAC, SRP, and others.

### 4.3. Shaping the Polypeptide Fate from the Ribosomal Surface

A detailed understanding of which cytosolic factors interact with the ribosome and where on the ribosome is emerging and SRP is now understood fairly well. It shares an overlapping binding site at the tunnel exit with NAC, which implies a competition of different factors for the egressing nascent chain. Unlike the far less abundant SRP, NAC actually exists in equimolar concentration to the ribosome and seems to scan nascent polypeptides by reaching into the exit tunnel [242,244]. This priority access enables NAC to orchestrate downstream factors. For example, the occupation of ribosomes by NAC prevents the unproductive, premature interaction between ribosomes synthesizing non-secretory proteins and the Sec61 complex [245]. On the other hand, SRP promotes the binding of ribosomes that synthesize secretory proteins with proper targeting signals to the Sec61 complex. In this regard, NAC has controversially been shown to modulate the binding affinity of SRP for non-secretory RNCs, thereby influencing the specificity of SRP-mediated ER targeting [217,246]. While one model excludes the simultaneous binding of NAC and SRP to the ribosome, a second model favors the co-binding of both factors near the exit tunnel with NAC acting as a negative regulator that selectively displaces parts of SRP from the exit port [247,248,249]. As a consequence, NAC hampers the efficient targeting of SRP to its receptor in the ER membrane and in the case of “signal sequence-less” RNCs and other factors may gain access to the cargo. Besides the regulation by NAC, the mechanisms for nascent chain- and mRNA-dependent recruitment (Figure 4) compensate for the low abundance of SRP and prioritize the occupation of secretory RNCs with SRP [191,222,234]. 

SGTA and the BAG6 complex, both attributed to the GET pathway (Figure 3), are known to transiently bind the mammalian ribosome [150,166,236]. Their exact location remains unclear, but the binding sites seem to be different from SRP as simultaneous and non-competitive interactions have been observed. In accordance with the hydrophobic character of all ER targeting signals, early recruitment to the ribosome in response to the translation of respective sequences appears as a common mechanism [150,236]. Interestingly, yeast lacks a homolog of BAG6 with consequences for the hierarchical organization of the GET pathway compared to mammalian cells. Specifically, Get4/5 (GET4/5 homolog) binds the yeast ribosome overlapping with SRP via the Get5 subunit [250]. Sgt2 (SGTA homolog) is recruited to the tunnel exit only in a subsequent step. Hence, competitive scanning of the ribosome by SRP and Get4/5 seems to drive the targeting fate in yeast, in accordance with the high prevalence of the post-translational transport mode. By contrast, the ribosomal contact in mammalian cells is mediated by the BAG6 subunit and SGTA also binds to the ribosome independently of the BAG6 complex (Figure 4). Questions regarding which factor catches the cargo first and why an additional component, BAG6, is located at the mammalian ribosome are currently unanswered. 

Furthermore, it has also been assumed that Snd1, the cytosolic component of the SND pathway in yeast, is recruited to the ribosome during mRNA translation [21,186]. Still, a homolog in mammalian cells has not yet been identified. How the corresponding clients with a central TMH are recognized and delivered to the mammalian ER thus remains to be clarified. In principle, several factors have the capacity to bind central targeting signals. First, SRP has been shown to be involved in all membrane protein targeting in yeast, including those with TMHs distant from the termini [125]. Second, SGTA may also bind internal TMHs, though it has been suggested that it assists SRP-mediated targeting by preventing aggregation, ubiquitination, and degradation [150].

Thus, the choice for a targeting pathway starts with the birth of the nascent chain inside the ribosome. The interplay between the elements of the cargo (mRNA motifs and codon choice, nascent chain sequence, and secondary structure), the ribosome (exit tunnel and surface), and the targeting factors influence the overall efficiency, specificity, and fidelity of the targeting process [248].

## 5. Different ER Protein Translocases Act as Membrane-Integrated Chaperones

Consistent with the multiplicity and complexity of targeting signals and targeting pathways described in the previous sections, recent findings in the field of ER protein import have also extended this concept to a small assortment of ER protein translocases. Different types and arrangements of protein translocases, sometimes acting in concert, manage the translocation or insertion of a dedicated subset of incoming polypeptides (Figure 5). The different multimeric protein complexes that catalyze the insertion of an unfolded polypeptide, nascent or full-length, behave in a way very similar to targeting factors acting as a temporary safe harbor. As such, a membrane-integrated protein translocase transiently shields segments of incoming cargo and thereby facilitates the partitioning of hydrophobic TMH into the ER membrane or the translocation of soluble domains into the ER lumen. Thus, incoming polypeptides are handed over from a soluble targeting factor to a membrane-integrated translocase, both of which provide a chaperone-like environment and prevent improper interactions and the aggregation of unfolded polypeptides. Here, we will summarize the central constituents of protein translocase complexes and refer the reader to excellent reviews found in this special issue on the “Mechanisms of ER Protein Import” as well as others [26,27,30,32].

### 5.1. The Sec61 Translocase—Director for SPs and the Majority of TMHs

The protein translocase embedded in the lipid bilayer of the ER described first was the heterotrimeric Sec61 complex (Figure 5). This universally conserved protein-conducting channel is composed of the pore-forming subunit Sec61α (Sec61p in yeast, SecY in bacteria) and two TA proteins, Sec61β and Sec61γ (Sbh1 and Ss1p in yeast, SecG and SecE in bacteria) [28,253,254,255]. Initially, a genetic screen of yeast mutants unable to translocate a SP-containing marker protein identified the yeast *SEC61* gene as an important constituent in the early stages of ER protein translocation [256,257]. Subsequently, the mammalian homolog of Sec61p was identified and its functional as well as structural characterization as the major polypeptide-conducting channel of the ER followed [79,258,259,260,261]. Complementing the functional conservation, many structural studies have also highlighted the conserved architecture of the Sec61 complex from bacteria and archaea to lower and higher eucaryotes [262,263,264,265,266,267,268]. In its closed conformation, the structural data depict the pore-forming Sec61α as a multi-spanning membrane protein with an hourglass-like configuration. Its funnels are oriented perpendicular to the plane of the ER membrane, thus facing the cytosol and the ER lumen [11]. The ten THMs are organized in a pseudo-symmetrical fashion with the first and last five TMHs forming an N-terminal as well as a C-terminal half of the molecule, respectively, and are connected by the short luminal loop5. The clamp-like movement of the Sec61 complex opens up the so-called lateral gate that is able to transiently harbor both types of sufficiently hydrophobic targeting signals, SPs and TMHs [80,264,269,270]. While SPs are usually cleaved off by a signal peptidase complex [271,272,273], TMHs are released either individually or as a small bundle into the ER membrane [112,274,275]. The lateral gate, which is mainly framed by the sterically adjacent TMH2 and TMH7 of Sec61α, also contributes to another important feature of the channel, the pore ring. The pore ring represents a constriction zone in the center of the Sec61 complex and consists of six bulky, hydrophobic residues, three of which are located in the lateral gate helices 2 and 7. Residues of the pore ring face towards the center and form a flexible gasket avoiding the excessive membrane permeability of small molecules and ions during the transport process. While certain exchanges of all six pore ring residues can be tolerated in yeast, the mutation of a single human pore ring residue such as V85D can have severe consequences causing primary antibody deficiency [276,277,278]. Based on the hourglass-like structural layout, the pore ring separates the two opposing funnels. In the idle state, the cytosolic funnel is water-filled, whereas the luminal funnel is occupied by another critical feature called the plug domain. The plug is formed by roughly 20 amino acids of the first luminal loop of the primary structure of Sec61α and supports the gating of the channel as a small, flexible helix. The transition from the closed to the open state of the Sec61 complex is associated with the widening of the pore ring and the displacement of the plug domain to form a conduit across the membrane [279,280,281]. However, some structural data also show snapshots of the transport process of TMH without major displacement of the plug [264]. Thus, the insertion of a TMH or a SP might have different requirements regarding the luminal and/or lateral opening of the Sec61 channel. Accordingly, additional membrane and soluble proteins, some of which will be described below, accompany the Sec61 complex and act as allosteric effectors that stimulate substrate-specific gating of the channel. Among different homologs of the Sec61 complex, the plug region shows low sequence conservation [31,282]. Similar to the pore ring, the plug shows a secondary role for function and cell viability in yeast, whereas a single dominant mutation in the human plug domain such as V67G can have severe consequences causing autosomal dominant tubulointerstitial kidney disease [282,283,284]. The polypeptides destined for transport by the Sec61 complex can be presented either co-translationally as a nascent chain in conjunction with the ribosome, or post-translationally as a full-length unfolded protein accompanied by a proper targeting factor [16,285]. Across all kingdoms of life, the Sec61 complex associates with different accessory factors to facilitate both variations of substrate transport, co- and post-translational [16,17,286,287,288,289]. Irremissible for substrate transport is the transition of the Sec61 complex from the closed to the open state. Unlike a simple lock, the opening of the Sec61 complex is a multifactorial, stepwise process mediated by the combination of a SP/TMH wedging itself into the lateral gate plus accessory factors such as the ribosome binding to a functionally conserved cytosolic docking port consisting of loops 6 and 8 [265,290,291]. In a mammalian setting, the cytosolic loops 6 and 8 of Sec61α are often referred to as the ribosome binding site. Yet, in light of existing structural and functional data from different organisms, this domain rather serves as a universal docking port for native (bacterial ATPase SecA, the translating ribosome, the ER membrane protein Sec63) as well as pseudo-native (heterologous anti-Sec61α F_ab_ fragment, autologous Sec61 molecules arising from crystal packing) ligands [80,86,262,264,265,266,267,292,293,294,295,296,297,298,299]. Those ligands are auxiliary factors that act as biocatalysts and lower the activation energy to support the opening and closing of the Sec61 complex for efficient protein transport [291]. Both termini of Sec61α face the cytosol and the somewhat longer N-terminus contributes as a binding spot for other regulatory factors such as calmodulin or post-translational modifications such as N-acetylation for the efficient gating of the Sec61α protein in mammals and yeast, respectively [300,301]. Although fewer data are available for the C-terminus, it may support ER retention or, as was recently shown for the C-terminus of yeast Sec61p, it stimulates ribosome binding and co-translational protein transport [302]. 

#### 5.1.1. Opening of the Sec61 Complex by Targeting Signals

The interplay of the structural key elements of the Sec61 complex together with targeting signals and accessory factors supports (i) the ER-specific entry and (ii) the proper topology of the transported substrates that are either fully translocated across, integrated via TMH into, or associated via a lipid-anchor with the ER membrane. Thus, hydrophobic targeting signals serve multiple purposes including targeting to as well as the gating of the Sec61 complex and may very well encode a post-translocation function for the topology and folding of the substrate itself. The latter is not surprising in the case of TMHs that are an integral part of the mature protein. Yet, some reports show the impact of cleavable SPs, or fragments thereof that are produced by the signal peptide peptidase that influence the folding of the downstream protein domain via specific chaperone recruitment or are used otherwise for the presentation of self-antigens via MHC class I [56,303,304,305,306,307,308]. Conclusions from research strategies such as in vitro protein import, photo-crosslinking studies, single Sec61 channel recordings from planar lipid bilayers experiments, and structural analyses of substrate-engaged Sec61 complexes have demonstrated the opening of the channel by SPs or TMHs of substrates, whereas SPs could be delivered co- or post-translationally [80,264,296,309,310,311,312,313,314,315,316,317,318]. To a certain degree, the process of opening the Sec61 complex by SPs and certain TMHs (excl. type III and type IV) resemble one another but the process has been studied best for a co-translationally directed SP, which will be outlined here. The binding of a translating ribosome to the universal docking port causes initial structural re-arrangements that prime the Sec61 complex and align the ribosomal exit tunnel with the nascent polypeptide plus SP on top of the cytosolic funnel. Moreover, the binding of the RNC causes the destabilization of a polar cluster within Sec61α which opens a crack in the cytosolic half of the lateral gate and exposes a single hydrophobic patch in the cytosolic funnel. The hydrophobic patch attracts the hydrophobic h-domain of an incoming SP and intercalates the targeting signal in the lateral gate while simultaneously supplanting the lateral gate of helix 2. Analogous to the placeholder helix that was described in client recognition by SRP above, helix 2 of Sec61α has been interpreted as a placeholder, setting the hydrophobicity threshold for productive interactions with targeting signals and productive channel gating. Some of the residues that form the polar cluster (two out of three amino acids), the hydrophobic patch (three out of four amino acids), and the pore ring (three out of six amino acids) all reside in the helices that form the lateral gate and further emphasize its importance as a critical structural element of the Sec61 complex [31]. Consequently, the intercalation of a targeting signal at the lateral gate has been extensively addressed in crosslinking reports [260,318,319,320,321,322] as well as computational analyses [323,324,325]. After engaging the Sec61 complex headfirst, biochemical assays, structural models, and coarse-grained modeling approaches have shown an inversion or a flip-turn of the SP as well as the corresponding TMH pendant of type II membrane proteins generating the N_cyto_/C_exo_ orientation of those targeting signals housed in the lateral gate upon re-orientation [80,97,98,325]. At this point, the channel is fully open with the pore ring widened and the plug domain displaced [80,290]. While the hydrophilic sequence elements downstream of the SP can enter the ER lumen, hydrophobic TMH can partition laterally into the membrane. During the transport process, the pore ring residues surround the polypeptide in transit to preserve the permeability barrier for other small molecules and ions [276,279,292]. 

#### 5.1.2. The Growing Family of Sec61 Complex-Associated Factors

This general concept of channel opening, substrate transport, and its topology determination is further influenced by many biophysical factors. For example, sequence composition as well as the length of the targeting signal and the following mature domain, the folding speed of flanking sequence elements, ribosomal translation speed, or the positive-inside rule all can affect translocation fidelity. In other words, the fruitful biogenesis of proteins that belong to the secretory pathway is not a “one molecule job”. It relies on information that is encoded by the nascent chain plus additional mRNA sequence elements and is initially decoded by the ribosome, targeting factors, and the Sec61 complex plus its transiently or permanently associated proteins, as well as the lipid environment of the membrane, and is subsequently complemented by further maturation and quality control elements [27,42,95,119,270,323,326,327,328,329,330,331,332]. Recent structural, proteomic, and reconstituted import studies have addressed the contribution of different substrate-specific factors that support the Sec61 complex during the gating of “imperfect” SPs, a few of which are shortly highlighted. The TRAP complex supports the Sec61 complex during the co-translational transport of SPs with an above-average content of helix-breaking glycine-plus-proline residues [89,333,334,335,336,337]. Using classical reconstitution approaches, the translocating chain-associated membrane protein 1 (TRAM1) has been found to aid the insertion of nascent chains into the Sec61 complex when their SP has a shorter than average N-region [319,336,338,339]. Further data have implied that TRAM1 may also assist protein import by making the lipid bilayer in the vicinity of the lateral gate of the Sec61 complex conducive for accepting targeting signals [340]. Alternatively, the Sec62/63 complex assists the substrate-specific post-translational or co-translational opening of the Sec61 complex [83,85,86,87,293,295,296,332,341]. Proteomic abundance analysis in mammalian cells has identified SPs with longer but fewer hydrophobic h-regions plus a lower C-region polarity that is dependent on the Sec62/63 module [342]. While the TRAP and Sec62/63 complex seem to assist mainly in the transport of SP-carrying precursor proteins, other trans-acting factors have recently been shown to support the Sec61 complex during the insertion of “imperfect” TMHs, especially those of polytopic membrane proteins. The trans-acting factors in question that await a more detailed functional characterization in the near future are the TMCO1 (transmembrane and coiled-coil domain-containing protein 1) and PAT complex [112,251,252,343]. Both these modules consist of multiple ER membrane proteins. The combination of cryo-EM and crosslink-MS data upon the affinity purification of TMCO1–ribosome complexes identified proteins of the ribosome and the Sec61 complex, and also the Nicalin–TMEM147–NOMO complex as well as CCDC47 (coiled-coil domain containing protein 47). Furthermore, the use of affinity-purified TMCO1–ribosome complexes allowed McGilvray et al. to sequence ribosome-associated mRNAs that are enriched at the TMCO1 translocon. While mRNAs encoding for soluble as well as membrane-spanning secretory proteins with up to three TMHs were underrepresented, those with four and more TMHs showed a strong enrichment [252]. The beauty of the TMCO1 translocon structure (PDB: 6W6L) is amplified by its ambiguity, simultaneously answering as well as opening up questions regarding the regulation of the ER protein translocase. The TMCO1 complex occupies much of the space that is reserved during the co-translational import by the oligosaccharyltransferase (OST) complex including contacts to rRNA helices H19 and H25 in the cytosol or the volume taken by OST subunit Stt3a in the ER lumen [294,344]. Either different translocon modules rapidly exchange and rearrange during the insertion of glycosylated multi-spanning proteins or these proteins are specifically glycosylated post-translocationally by the OST paralog acting independently of the Sec61 complex [294,345,346]. Similar to OST and TRAP, the TMCO1 complex lingers on the “back side” of the Sec61 complex opposite the lateral gate as a second belt in addition to Sec61γ. This positioning makes it hard to envision a direct contact of a TMH with the lipid-filled cavity provided by the TMCO1 complex on the back side of Sec61. Yet, some of the unaccounted density of a TMH in the OST structure shown by Braunger et al. could represent the imperfect TMH1 of the bovine opsin substrate used for purification [294]. Eventually, this TMH1 meanders around the outskirts of the Sec61 complex to the back side to find the lipid-filled cavity of the TMCO1 complex. On the other hand, this positioning grants the TMEM147 subunit of the TMCO1 complex easy access to the hinge region (luminal loop 5) connecting the N- and C-terminal halves of Sec61α, thereby supporting the lateral opening and release of a TMH. The long α-helical arm of CCDC47 (Asn399-Lys474) in the cytosol moves in close proximity to the universal docking port (cytosolic loops 6 and 8) of Sec61α and the ribosomal protein uL22 at the ribosomal exit tunnel. Hence, CCDC47 might impact the opening of Sec61α, the directionality of an incoming TMH plus its flanking regions, or ribosomal translation. Last, the large luminal portion of Nicalin could aid translocated loops and domains of polytopic membrane proteins or, contrariwise, shield these domains from the luminal chaperone BiP from either ratcheting them or prematurely closing the Sec61 complex [347,348]. Overall, the players of the TMCO1 complex are arranged in a similar way to a relay system with CCDC47–TMCO1–TMEM147–Nicalin mainly contacting the ribosome-Sec61 complex ER lumen, respectively [252]. Interestingly, an independent study also identified CCDC47 with a functional link to membrane protein biogenesis [251]. The heterodimeric complex consisting of CCDC47 and Asterix (Pat-10) was termed PAT, short for proteins associated with the ER translocon. Using site-specific crosslinking and stability analyses of bicistronic reporter proteins via flow cytometry, the PAT complex was shown to facilitate the biogenesis of polytopic membrane proteins with imperfect, hydrophilic TMHs after such TMHs were inserted into the lipid bilayer [111,112,251]. Consequently, the PAT complex is considered to be an intramembrane chaperone. Via the shared subunit CCDC47, the PAT complex could act in conjunction with the TMCO1/Sec61 translocon. This “ménage à trois” creates an operational protein conducting channel whose active center (Sec61 complex) is supported by an allosteric effector (TMCO1 complex) and a folding assistant (PAT complex) for the proper biogenesis of polytopic membrane proteins with imperfect TMH (Figure 5). Interestingly, in this setup, at least one subunit of each module (Sec61α, TMCO1, CCDC47) in the Sec61–TMCO1–PAT complex has been shown to be involved in calcium homeostasis of the ER, highlighting a tight connection between protein transport, membrane permeability, and calcium signaling [349,350,351].

Different lines of evidence based on immunodepletion or the chemical inhibition of the Sec61 complex have revealed the limitations of the Sec61-centric translocases for the insertion of certain bi- and polytopic membrane proteins [99,352,353,354]. Most notably, membrane proteins carrying a type III or type IV TMH are more resistant to the inactivation of the Sec61 complex and instead can be inserted by the EMC or GET complexes. Similar to TMCO1, the EMC and GET complex harbor subunits that also belong to the Oxa1 superfamily [29,355].

### 5.2. The GET1/2 Complex—Post-Translational Machine for Type IV TMH of TA Proteins

A priori, the absence of an SP and the C-terminal localization of a TMH defined as type IV requires a specialized, i.e., a post-translational, targeting and insertion mechanism. Many components of this pathway are abbreviated by the acronym GET, guided entry of TA proteins [203]. TA polypeptides are fully synthesized and released from the ribosome while their type IV TMH is recognized by a pre-targeting complex and handed over to GET3 (see Section 3.2). Although some TA proteins have been reported to insert spontaneously into protein-free liposomes in vitro, the macromolecular crowding and the presence of different organellar target membranes under cellular conditions require an insertase for TA proteins [117,118,356]. In yeast and mammals, the ER proteins GET1 and GET2 form a complex (Figure 5) that first serves as a membrane-integrated receptor for incoming GET3–TA protein cargo and subsequently as an insertase for the C-terminal TMH [19,154,357,358,359]. Overall, the fundamentals of the GET system for the targeting and insertion of TA proteins resemble those of the Sec61 complex for secretory proteins. The four-step mechanism includes (i) the recognition of targeting signals by a targeting factor, (ii) the transfer of the cargo complex to an ER-localized receptor, (iii) the release of the cargo from the targeting factor for cargo insertion by a translocase, and (iv) the recycling of the targeting factor.

#### The GET1/2 Duality—Receptor and Insertase Function

Studies addressing the insertion mechanism of TA proteins by the GET1/2 complex point at a monomeric or homodimeric assembly of the GET1-GET2 heterodimer as a minimal functional unit [359,360]. Irrespective of the oligomerization, the GET1/2 complex attracts the active GET3 dimer that delivers a single TA protein. Both proteins, GET1 and GET2, have three TMHs and an extended cytoplasmic domain capable of binding to GET3. Thus, they likely cooperate in the targeting and release of a TA protein. The current model suggests that the long, unstructured N-terminal tether of GET2 captures the GET3–TA protein complex and brings it into proximity to the coiled-coil domain of GET1. In contrast to the tether of GET2, GET1’s coiled-coil domain triggers a conformational change in the GET3–TA protein complex triggering the release of the cargo close to the insertase domain, which is ascribed to the TMHs of the GET1/2 complex and includes the conically shaped hydrophilic groove formed by the three TMHs of GET1 [153,154,358,359,361]. In contrast to the aqueous pore of the Sec61 complex traversing the entire ER membrane, the hydrophilic groove of the GET1/2 complex forms a discontinuous vestibule or “hemi-channel” that grants access to the cytosolic part of the membrane, but is sealed on the lumenal leaflet by an additional α-helix running parallel to the plane of the membrane [29]. Each one of the three TMHs of human GET1 carries at least one positively charged residue that rests in the cytosolic facing half of each TMH. Therefore, the hydrophilic vestibule might support both the membrane passage of the short, hydrophilic tails of the C-terminal of the type IV TMH and the “decoding” of the positive inside rule for proper TA protein topology by blocking the translocation of positive charges [93,330]. Membrane integration of a TA protein might further be stimulated by GET2-mediated membrane thinning. Unlike the archetypical 21 amino acid TMH running perpendicular through the membrane (Figure 2B), all three TMHs of human GET2 are shorter, comprising only 18–19 amino acids. The short TMHs of GET2 cause a local bilayer destabilization that promotes the integration of a TA protein. After the release and integration of the TA protein, the cargo- and nucleotide-free GET3 is recycled. The cytosolic GET components GET4/5 promote the rebinding of ATP in the empty nucleotide-binding pockets of the GET3 dimer and its dissociation from the GET1/2 complex [362]. Although both organisms, yeast and mouse, encode the TA proteins essential for viability, the deletion of the *GET* genes is tolerated by the yeast cells, whereas the constitutive genetic ablation of *GET3* or *GET2* causes embryonic lethality in mice [19,127,363]. The conditional knockout of GET components in certain tissues is viable in mice, but shows abnormalities sometimes related to improper TA protein biogenesis [129,130,364,365,366]. One assumption, mainly driven by the viability of GET-deficient yeast cells, was the existence of an alternative insertion machinery for TA proteins. 

### 5.3. The EMC—Emcee for Type III, Type IV, and Charge-Containing TMHs

Similar to the central components of the GET pathway, the initial discovery of the EMC was achieved by high-throughput screening in yeast, where genetic interactions revealed their functional relationships [367]. The functional conservation and importance of EMC for the proper protein biogenesis of membrane proteins was subsequently demonstrated in many organisms including worms, zebrafish, flies, mice, and human cells [368,369,370,371,372]. Depending on the organism, EMC harbors 8–10 subunits, with seven of them being ER membrane proteins that are accompanied by up to three cytosolic proteins (EMC2/8/9) as is the case for the human EMC [373]. Detailed functional analyses based on reconstitution, proteomics, and proximity-specific ribosome profiling approaches narrowed down the substrate spectrum of EMC. This multimeric assembly preferentially supports the biogenesis of two classes of membrane proteins: TA proteins that are skipped by or are not ideally suited for GET3 and polytopic membrane proteins that either start with a type III TMH or carry charged and aromatic residues in the TMHs. All these types of TMH are difficult to handle for the Sec61 complex (Figure 5). In addition, mutational analyses of the TMHs of EMC substrates further supported these findings [99,157,372,374]. Considering the client spectrum, it appears that EMC can act in two different modes either as post-translational insertase for TA proteins or as co-translational insertase plus intra-membrane chaperone for individual TMHs of polytopic membrane proteins [157,251,375]. Similarly, the structure-guided mutational analyses of yeast and human EMC subunits have suggested the multifunctionality of this complex during the biogenesis of different membrane proteins [376,377,378]. Within the membrane-embedded core of the EMC, Miller-Vedam et al. defined two cavities, a lipid-filled and a gated one, on opposite sides of the transmembrane core and with different functionalities. The gated cavity is lined by portions of EMC3, EMC4, and EMC6 and seems to provide the actual conduit for the insertion of the terminal TMHs from either TA proteins or polytopic membrane proteins. This cavity resembles the hydrophilic vestibule of GET1/2 that promotes the access of TMHs to the lipid bilayer. Similar to TMCO1 and GET1, EMC3 also belongs to the class of Oxa1 superfamily members providing an evolutionary perspective on the conserved functional principle of these translocase subunits [29,355]. Overall, multiple cryo-EM structures of EMC and GET1/2 propose a similar mechanism for the integration of TMHs into the ER membrane. As described above for GET1/2, the EMC seems to adhere to the translocation-stimulating concept of the Oxa1 superfamily and employs local membrane thinning and a hydrophilic vestibule as a hemi-channel too [29,375,376,377,378]. With the limited size and discontinuity of the hemi-channel, one can appreciate the limited ability of EMC and GET1/2 to translocate larger flanking domains and their preference for N-terminal type III or C-terminal type IV TMHs, usually requiring the translocation of smaller flanking domains [26]. The function of the lipid-filled cavity that is built by portions of EMC1/3/5/6, is less clearly defined and more pleiotropic [376]. However, as this cavity shows a uniformly hydrophobic surface and appears to be accessible from the membrane or the ER lumen, it may provide a proper space for the intra-membrane chaperone and holdase function of the EMC.

#### The Intra-Membrane Handover between EMC and Sec61

A major difference between the Sec61 complex capable of opening a continuous, aqueous pore in comparison to the hemi-channels of EMC and GET1/2 is the translocation of larger domains into the ER lumen. Only the Sec61 complex seems to be efficient in transporting larger polypeptide stretches across the membrane. In the case of the Sec61 complex, this also includes soluble, secretory proteins that seem to be exclusively handled by this protein translocase. In contrast, membrane proteins with an initial type III stop-transfer or a type IV TMH that both require the translocation of shorter domains into the ER lumen are preferentially handled by EMC and GET1/2, respectively. Considering the size of EMC, ribosomes, SRP, and the Sec61 complex, a major question that might soon find an answer relates to the resulting steric constraints during the co-translational insertion of polytopic membrane proteins by EMC. This activity might entail the cooperation of EMC with the Sec61 complex and the RNC and require a uni- or bidirectional handover between the translocases. However, once the ribosome binds with high affinity to the Sec61 complex, the minimum distance between the lateral gate of Sec61α and the hemi-channel of EMC is approximately 110 Å [375]. Therefore, it appears rather unlikely that EMC captures TMHs at the Sec61 lateral gate and more likely that EMC mediates the insertion of TMHs before ribosomes bind to the Sec61 complex. This cooperation between two translocases, or intra-membrane chaperones, resembles the handover of polypeptide cargo from an upstream to a downstream targeting factor partially driven by increasing affinity, a concept that was discussed above.

The transport of polypeptides into the ER lumen or the ER membrane is usually followed by folding, modification, and eventually the assembly of proteins to achieve a native conformation. Similar to the targeting and translocation process, polypeptides that enter the ER encounter several molecular chaperones and co-chaperones that in this case support the folding and quality control process. Intriguing details about the impact of such factors on protein biogenesis have been reviewed previously and are not further discussed here [379,380,381,382].

## 6. Disease-Causing Mutations of Targeting and Translocation Components

As a result of the complex orchestration of the protein targeting network, the misfunctions of single players disrupt its sensitive balance and mutations of different protagonists have led to a broad spectrum of pathological phenomena. Disease-causing mutations have been identified in pivotal components in many of the protein targeting and transport pathways discussed before (Figure 6).

### 6.1. Disease Associations of Protein Targeting Factors

Beginning with the soluble components of the targeting pathways, SRP72, one of the six proteins of the SRP, forms a complex with SRP68 and binds the 7S RNA to guide the pre-SRP complex from the nucleus to the cytosol [140,383,384]. Two mutations of the *SRP72* gene are known to cause the pathological phenotype of aplastic anemia, a developmental defect causing maturation defects of blood cells in the bone marrow that can progress into acute myeloid leukemia. The dominant SRP72 mutations result in a frame-shift causing the appearance of a premature stop codon (p.Thr355Lysfs^∗^19) or in an amino acid exchange (p.Arg207His). While the truncated SRP72 is ineffective in binding the 7S RNA component, it has been speculated that the mutant SRP72-R207H protein binds less efficiently to ribosomal proteins, SRP68, or SRP54 [385,386]. Similarly, SRP54 appears to be disease-related. Its function has been described as an RNA binding protein and it mediates the interaction with the SRP receptor in the ER membrane [387,388,389]. Autosomal dominant mutations in *SRP54* are known from three different patients and result in neutropenia with similarities to the Shwachman-Diamond Syndrome. The three de novo missense variants of SRP54 all affect the conserved residues of the GTPase domain known to be critical for GTP and receptor binding. In two of the three patients, their neutropenia, due to the *SRP54* mutation, was also accompanied by an exocrine pancreatic insufficiency [390]. Of note, although not based on mutations in the *SRP54* gene, auto-antibodies directed against the SRP54 protein are considered as diagnostic biomarkers as well as pathogenic agents driving the progression of immune-mediated necrotizing myopathy, a muscle-specific autoimmune disease [391,392]. Mutations in the *SRP72* and *SRP54* genes demonstrate the importance of efficient protein secretion for the professional secretory cell types of the immune system and pancreas (Figure 6). Regarding the subunits of the GET pre-targeting complex BAG6 and co-chaperone SGTA, we wish to refer readers to a recent review on the roles of these cytosolic quality control proteins in disease [393]. Briefly, evidence from different cohorts and meta-analyses for the *BAG6* gene has linked certain single nucleotide polymorphisms to an increased risk for lung cancer and osteoarthritis [394,395,396,397]. Further experimental data have also suggested that the BAG6 protein is involved in male infertility and autoimmune disease [393]. The co-chaperone SGTA has been discussed in the context of different types of cancers including esophageal squamous cell carcinoma, breast cancer, and lung cancer. In all three cases, the SGTA protein showed an elevated abundance in cancerous tissue samples and was correlated with shorter survival rates [398,399,400]. Although the underlying mechanism is still unclear, these studies have reported on the impact of SGTA on cell proliferation and cell cycle progression. A single nucleotide polymorphism of the *SGTA* gene is associated with polycystic ovary syndrome, an endocrine disorder causing a hormonal imbalance in women characterized by increased levels of androgens [401]. Women suffering from the syndrome are more likely to develop endometrial cancer.

### 6.2. Disease Associations of Receptor and Protein Translocase Subunits

Aside from the protein targeting components, mutations have also been identified in genes encoding for subunits of the ER protein translocation machines (Figure 6). Pathologic functions and phenotypes connected with the Sec61 protein and its interaction partners Sec62 and Sec63 have recently been reviewed elsewhere [402,403]. Related diseases that arise from mutations or the overabundance of these proteins are diverse and include different types of cancer (Sec61γ, Sec62, Sec63), autosomal dominant polycystic liver disease (Sec61β, Sec63) as well as common variable immunodeficiency, neutropenia, and autosomal dominant tubulointerstitial kidney disease (Sec61α) [276,283,403,404,405,406,407,408]. Mutations in other allosteric effectors of the Sec61 channel such as the TRAP and OST complex are also the reason for severe pathologies based on disorders of N-glycosylation [409,410]. Components of the recently described TMCO1 and PAT complex are also disease associated upon mutation. As such, different autosomal recessive nonsense-mutations that entail the premature translational termination of TMCO1 (e.g., p.Ser47*, p.Arg87*, p.Ser98*, p.Arg114, etc.) have been sequenced that cause cerebro-facio-thoracic dysplasia. This multisystem developmental disorder causes intellectual disability, facial dysmorphism (a wide and short skull, highly arched eyebrows, widely spaced eyes), as well as abnormalities of the ribs and spinal bones [411,412,413,414]. The frequent malformation of bones in this context raises the question of whether the underlying pathogenic mechanism involves the functions of TMCO1 related to protein transport and/or calcium homeostasis. CCDC47, which appears to be part of the TMCO1 as well as of the PAT complex, has been associated with tricho-hepato-neuro-developmental syndrome. This multisystem disorder is characterized by woolly hair, pruritus (itching), hepatic dysfunction, general dysmorphic features, and developmental delay. Homozygous and compound heterozygous recessive alleles of the *CCDC47* gene can result in the absence of the functional CCDC47 protein. Its loss in patient fibroblasts has been shown to decrease calcium storage in the ER and insufficient calcium refilling during store-operated calcium entry [415]. While the experimental evidence demonstrates the dysregulation of calcium homeostasis upon loss of CCDC47, it is unclear if the patients with tricho-hepato-neuro-developmental syndrome also suffer from distinct abnormalities of protein transport related to the translocase and chaperone function of the TMCO1 and PAT complexes. For the other subunit of the PAT complex, Asterix (WDR83OS), a recessive variant has been reported to cause an unspecified syndrome with intractable itching, facial dysmorphia, microcephalus, hypercholanemia, short stature, and intellectual disability. Comparing these features with the ones from the loss of CCDC47, some similarities such as itching, hepatic dysfunction, dysmorphic features, and developmental delay are evident. The phenotypic overlap upon loss of either PAT complex component speaks in favor of the genetic and functional interaction of the two genes and the encoded wild-type proteins CCDC47 and Asterix.

Mutations in the subunits of the EMC are also related or causative for different diseases including neurological disorders or cancer. For instance, a defect of the EMC1 protein is related to visual disorders, craniofacial abnormalities, and epilepsy [416,417].A recessive loss of function related to the mutation of the *EMC10* gene has been reported in patients with intellectual disabilities and developmental delay [418]. Interestingly, the overexpression of either *EMC6* or *EMC10* has been reported to provide anti-tumor activity in glioblastoma cells. The tumor-suppressor activity upon the overexpression of those EMC subunits has been attributed to changes in gene expression that slow down cell proliferation, cell cycle progression, and tumor invasiveness or changes in signaling that increase autophagic flux via the inactivation of the mTOR pathway [419,420,421]. 

The majority of components of the GET pathway and its function are conserved from yeast to plants and mammals [153,422,423]. With regard to illness, the genomic locus of the *GET1* gene has been mapped to the congenital heart disease region of human chromosome 21 [424]. Yet, a clear correlation between *GET1* and its role in the development of cardiac defects and congenital heart disease has not been established in humans. However, direct evidence for this association comes from non-human studies. It has been shown that after the depletion of the GET1 protein in embryos of Xenopus frogs and Medaka fish, the development of their heart was initiated, but its morphogenesis could not be finalized [425,426]. A more targeted approach testing the tissue-specific knockout of the *GET1* gene in cardiomyocytes of mice reported hepatic damage and fibrosis, but no heart-specific phenotype was observed either [129]. Regarding the GET2 protein, studies on human cancer cell lines have presented evidence for an association of GET2 with skin and breast tumors due to the effects on either the calcium or prolactin receptor signaling pathways [427,428]. Yet, mechanistic details related to the biogenesis of TA proteins are scarce in these studies. Work on Myc-induced B-cell lymphoma cells from mice has described the relevance of GET2, in particular its C-terminal GET1 binding domain, for the survival and mitotic progression of lymphoma cells. In this case, the oncogenic potential of GET2 was shown to be independent of its TA protein insertion function [177]. Similarly, no mutation providing evidence for a gene-disease association with clinical significance of the GET3 protein is listed in the OMIM (www.omim.org, accessed 26 October 2021) or gnomAD (https://gnomad.broadinstitute.org, accessed 26 October 2021) database. Thus, recessive mutations that affect the functionality of the components of the GET pathway could eventually be compensated for and such compensation might suppress any disease phenotype. However, homozygosity (or compound heterozygosity) of mutant alleles might be detrimental as demonstrated by mouse models for *GET1-3* and the embryonic lethality of the constitutive knockouts [127,363,429].

Less is known about the SND protein targeting pathway in mammalian cells and its correlation with diseases or in vivo malfunctions. Considering that the knockout of SND components in yeast leads to defects in carboxypeptidase Y maturation (Snd2) or phosphate regulation and connectivity between nuclear envelope and vacuole (Snd3), there could be a defect in mammalian cells upon the loss of SND [430,431,432]. In 2019, Talbot et al. generated a hSnd2 knockout in cultivated cells which resulted in the reduced expression of some polytopic membrane proteins including TRPC6 and KCNN4, two ion channels of the plasma membrane [185]. Maybe a connection between the human SND pathway and diseases might be based on those substrates of the pathway. As well as gain-of-function mutations, loss-of-function mutations of the *TRPC6* gene that can cause chronic kidney disease affecting the glomerulus have also been described. The corresponding disease is defined by a focal and segmental glomerulosclerosis phenotype [433,434,435]. The other SND substrate, KCNN4, has been associated with congenital hemolytic anemia. *KCNN4* mutations that were predicted to be deleterious perturb the cation permeability of cells and can lead to primary erythrocyte dehydration. The dysregulated volume homeostasis might affect the deformation capacity of erythrocytes and microcirculation which together could play a role as a pathogenic mechanism for hemolytic anemia [436,437]. However, direct disease associations of the SND pathway remain to be established.

All in all, the diseases caused by mutations of cytosolic or membrane-located targeting and transport factors show a broad phenotypic variance and affect different organs and tissues. This might be a result of the network organization of the different pathways and their central function in the biosynthesis of membrane or secreted proteins. Along the same line, defects of individual ER targeting, and translocation components might affect the efficient biogenesis of a limited client spectrum. The total loss or reduced abundance of tissue-specific proteins could also cause or contribute to a localized disease phenotype. As is the case of the widely present Sec61α protein, the majority of analyzed Sec61α point mutations result in distinct pathological phenotypes in different organs or tissues and eventually reflect the substrate-specific requirements found in the affected body part [276,283,404,438,439].

## 7. Conclusions and Perspectives

The mechanism of protein translocation through eucaryotic membranes is now more complicated than the mechanism described five decades ago. The original idea of a single targeting factor, a single receptor, and a single protein translocase has been massively expanded during the last years and now a variety of targeting routes and translocases dedicated to the transport of different polypeptide precursors has been shown. For certain proteins, it is clear how they are conducted to the ER, but some proteins can be delivered in alternative ways. The circumstance that allocates a polypeptide at a given time to use a certain transport pathway remains unclear.

The fate of the nascent polypeptide chain is majorly influenced by the type of targeting signal and hydrophobicity of the latter. It is not only the hydrophobicity that has an impact on the targeting mechanism, associated proteins and the local concentration of receptor components can all influence the transport destiny. It seems that different pathways can complement each other depending on the cellular environment. Cells may adjust to the different circumstances and fine-tune all the pathways to achieve the best-desired performance in terms of protein transport into the ER.

Clearly, more research is required to clear the possibilities and modes for protein translocation across the ER membrane. State-of-the-art structural and functional methods will unravel the enigmatic mechanisms of the different targeting routes and translocases. Even though we can find specific features for particular substrates, as always, there can be exceptions.

## Figures and Tables

**Figure 1 ijms-23-00143-f001:**
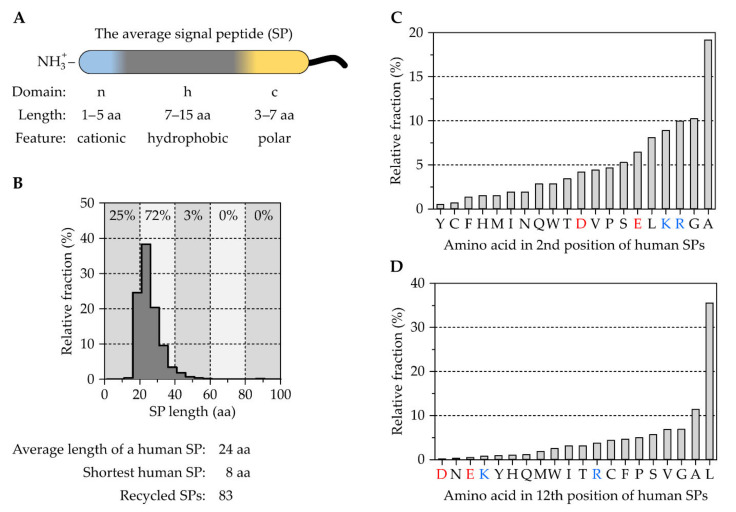
Features of human signal peptides and the amino acid distribution in key positions. (**A**) The tripartite segmentation of a cleavable signal peptide (SP) found at the N-terminus (NH_3_^+^) of secretory proteins. Characteristic attributes of the three sub-domains are given below. (**B**) A histogram showing the SP length distribution of 3584 human SP annotated at Uniprot. Numbers at the top of the diagram represent frequencies of the highlighted area. Almost three-quarters of SPs are 21–40 amino acids (aa) in length. (**C**,**D**) Histograms showing the frequencies of the twenty proteinogenic amino acids in the second (**C**) of twelve (**D**) positions of the same human SPs considered in (**B**) ordered from the lowest to the highest fraction. Amino acids are listed on the x-axis according to the one-letter code. The acidic amino acids aspartate (D) and glutamate (E) are labeled in red and the basic amino acids arginine (R) and lysine (K) in blue.

**Figure 2 ijms-23-00143-f002:**
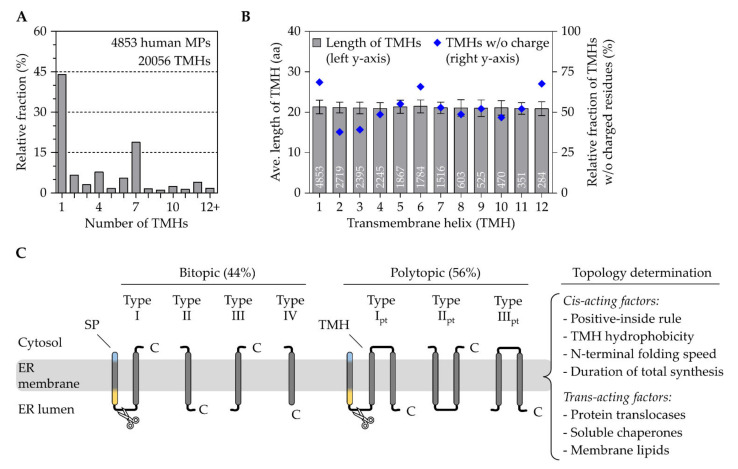
Length, charge, and topology of transmembrane helices of human bi- and polytopic membrane proteins. (**A**) The histogram shows the relative fraction of bitopic membrane proteins (MPs) with one transmembrane helix (TMH) and polytopic MPs with two or more TMHs. The data are based on the 20,056 TMHs of 4853 human MPs annotated at Uniprot. (**B**) Grey bars show the average length (left y-axis) with standard deviation of all first, second, and later TMHs till the twelfth TMH. White numbers at the bottom of the bar indicate the number of TMHs found in the human MPs. The blue diamond represents the relative fraction of the same TMHs that do not carry any charged residues (right y-axis). (**C**) Classification of bi- and polytopic (_pt_) membrane proteins based on topology, presence of a signal peptide (SP), and localization of a TMH within the primary structure. SPs are colored according to Figure 1 and their cleavage by the signal peptidase is indicated by scissors. For orientation purposes, the C-terminus (**C**) of each representative is indicated. Black lines indicate domains up- and downstream of a TMH. Cis- and trans-acting factors that influence the final topology of MPs are mentioned on the right and are further described in the text. aa, amino acids; ave., average; *w*/*o*, without.

**Figure 3 ijms-23-00143-f003:**
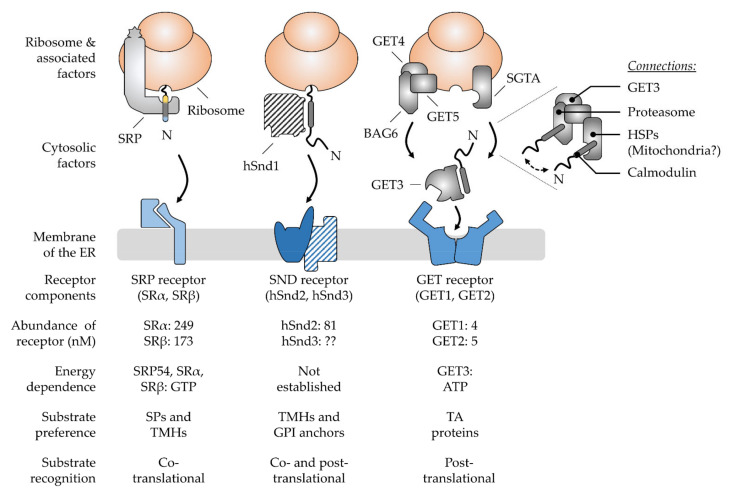
Major components and hallmarks of the mammalian SRP, SND, and GET targeting pathways. The top half shows a graphical output of the major components that shape the three targeting pathways SRP, SND, and GET. The dotted lines indicate a zoomed-in view of the BAG6 pre-targeting complex cooperating with SGTA and other cellular components. The double-headed arrow suggests the cycling of TA proteins between the two chaperones SGTA and BAG6. The bottom half summarizes some of the key features that differentiate the pathways from each other. Ribosome-associated and cytosolic targeting components are shown in grey colors and the cognate membrane receptors in shades of blue. Components (hSnd1, hSnd3) shown with a hatched color fill have not yet been identified in higher eucaryotes. Their existence is based on findings from yeast and the conserved nature of targeting machineries [21]. Abundance values for the receptor components are based on the quantitative mass spectrometry of mammalian cells [132]. Please note, there is considerable controversy about the abundance of GET1 and GET2 with some sources finding GET2 in four- to sevenfold molar excess over GET1 [133,134]. The N-terminus (N) of newly synthesized polypeptides is shown to accentuate the positioning of signal peptides (SPs) and transmembrane helices (TMHs) of different types of cargos, including tail-anchored (TA) proteins and glycosylphosphatidylinositol (GPI)-anchored proteins. BAG6, BCL2-associated athanogene 6; GET, guided entry of TA proteins; HSPs, heat shock proteins; SGTA, small glutamine rich tetratricopeptide repeat co-chaperone alpha; SND, SRP-independent; SR, SRP receptor; SRP, signal recognition particle.

**Figure 4 ijms-23-00143-f004:**
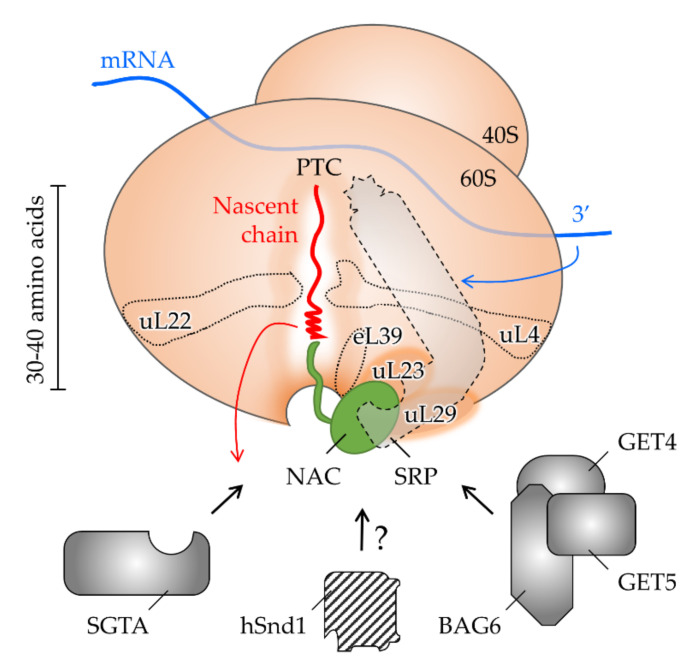
The impact of ribosome-binding proteins and the ribosomal exit tunnel geometry on the nascent chain and targeting decision. The nascent chain (red line) is born at the peptidyl transferase center (PTC) and folds in the vestibule of the lower ribosomal tunnel after passing the constriction sites build by the universally conserved ribosomal proteins uL22 and uL4 (dotted lines). Early recruitment of targeting factors involves the mRNA (blue line) and the presence and folding of the nascent chain while still residing inside the tunnel (blue and red arrow, respectively). Ribosomal surface proteins such as uL23 and uL29 (orange) which build common docking sites are mapped next to the exit port. Moreover, eL39 (dotted ellipse) that lines the interior of the ribosomal tunnel and spans to its surface might contact the nascent chain and putative targeting factors. Similarly, NAC reaches deep into the tunnel where it scans nascent chains to coordinate between the various factors competing for the emerging nascent chain (black arrows). Accordingly, NAC may regulate the targeting priority of targeting factors including SRP, the BAG6 complex, SGTA, or the putative hSnd1 (hatched color fill) to ensure the targeting of different types of cargo. Note that NAC and SRP were mapped at overlapping sites next to the tunnel, while the location of alternative factors remains unknown. BAG6, BCL2-associated athanogene 6; GET, guided entry of TA proteins; NAC, nascent polypeptide-associated complex; SGTA, small glutamine-rich tetratricopeptide repeat co-chaperone alpha; SND, SRP-independent; SRP, signal recognition particle.

**Figure 5 ijms-23-00143-f005:**
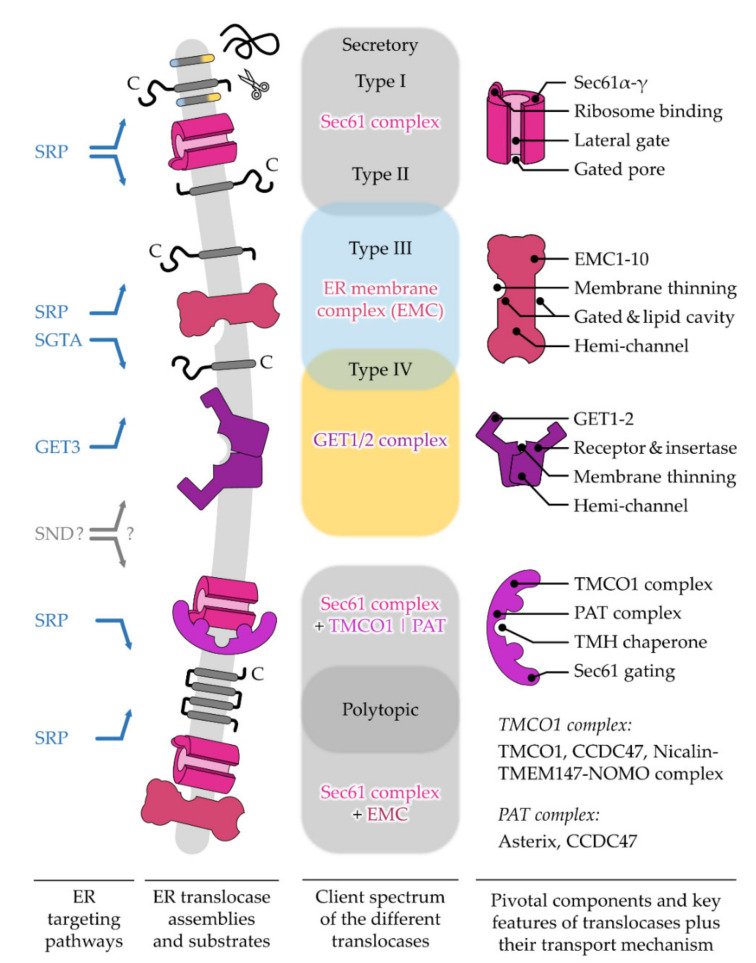
Features of protein translocation machines and their substrate spectrum. Depicted from the left to right are targeting pathways (blue), membrane-integrated protein translocation machines (pink, purple), their favored types of clients (boxes), and some key features for the central components of the protein translocation machines. Apart from GET1/2, the membrane-anchored receptor components for the targeting pathways are not shown. The scarcely characterized SND pathway (grey) as an alternative targeting route for a subset of GPI-anchored, tail-anchored, short secretory, and polytopic membrane proteins is indicated once. If SND delivers substrates either to a preferred ER translocase or a different one, or if its membrane-embedded component(s) might perform a dual function as receptor plus insertase similar to the GET1/2 complex remains to be seen. The auxiliary TMCO1 and PAT complexes are presented as one assembly. Of note, the PAT complex that was also described as a stand-alone intramembrane chaperone complex, is a heterodimer comprised of the indicated proteins CCDC47 and Asterix [251]. The TMCO1 containing translocon comprises the Sec61 channel and five accessory factors: TMCO1, CCDC47, and the Nicalin–TMEM147–NOMO complex [252]. For reasons of simplicity, we refer to those five accessory factors as “TMCO1 complex”. The PAT and TMCO1 complex share at least one subunit, CCDC47. Further details of the TMCO1–PAT–Sec61 assembly and how they might act in concert to form an operational protein translocase can be found in the text. Signal peptides and transmembrane helices are integrated into the ER membrane and classified and colored according to Figure 1 and Figure 2. The cleavage of signal peptides for secretory and type I membrane proteins is indicated (scissors and cleaved signal peptides are shown) as is the C-terminus (C) for each type of membrane protein. The spectra of substrates preferentially handled by assemblies that entail the Sec61 complexes are represented by grey boxes. Substrates handled by the ER membrane complex (EMC) and the GET1/2 complex are highlighted by light blue and yellow boxes, respectively. Overlapping boxes depict overlaps in the substrate range. CCDC47, coiled-coil domain containing protein 47; NOMO, nodal modulator protein; TMCO1, transmembrane and coiled-coil domain-containing protein 1; TMEM147, transmembrane protein 147.

**Figure 6 ijms-23-00143-f006:**
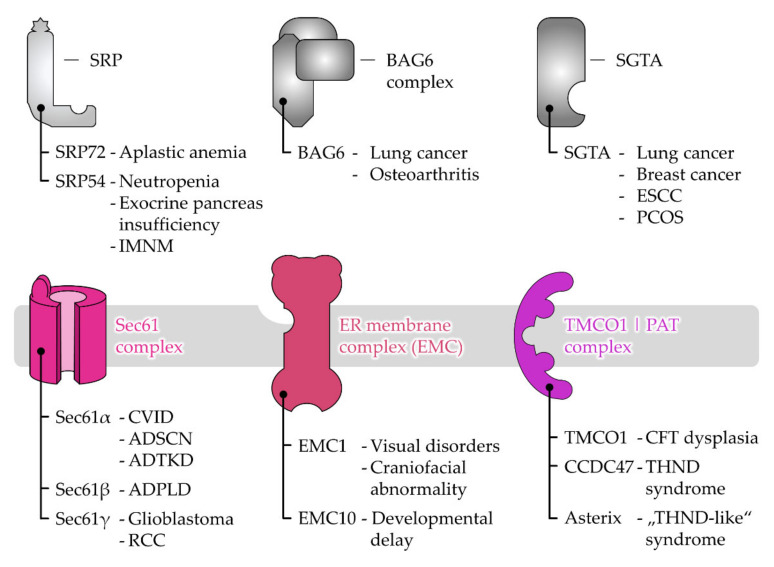
Disease associations of selected targeting and translocation components. The cartoon summarizes disease associations for critical subunits of targeting factors and translocation machines. Further details can be found in the text. ADPLD, autosomal dominant polycystic liver disease; ADSCN, autosomal dominant severe congenital neutropenia; ADTKD, autosomal dominant tubulo-interstitial kidney disease; CFT dysplasia, cerebro-facio-thoracic dysplasia; CVID, common variable immunodeficiency, ESCC, esophageal squamous cell carcinoma; IMNM, immune-mediated necrotizing myopathy; PCOS, polycystic ovary syndrome; RCC, renal cell carcinoma; THND syndrome, tricho-hepato-neuro-developmental syndrome.

## Data Availability

Sequences of human signal peptides and membrane proteins used for the analyses of Figure 1 and Figure 2 were extracted from www.uniprot.org. The protein abundance of receptor subunits reported in Figure 3 were taken from Hein et al. [132].

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
