# Peer review of "The Molecular Biodiversity of Protein Targeting and Protein Transport Related to the Endoplasmic Reticulum"

_ijms, 2021, doi:10.3390/ijms23010143_

Round 1
Reviewer 1 Report
I enjoyed reading this review. Usually I prefer shorter, more focused reviews, but here I have to congratulate the authors for their huge effort. I recommend publication as is. I only have 4 corrections.
1 - Please include a glossary/list of abbreviations (SRP, GET, SND, EMC, NAC,...)
2- In my pdf, text was deleted between l852 and l858
3- l1183 - "Emcee"
4 - Not all references have a DOI link
Author Response
Comments and Suggestions for Authors
I enjoyed reading this review. Usually I prefer shorter, more focused reviews, but here I have to congratulate the authors for their huge effort. I recommend publication as is. I only have 4 corrections.
We thank reviewer 1 for the positive feedback and the appreciation of our work. We are grateful for the structural suggestions and have addressed the points raised by the reviewer for the resubmitted version.
1 - Please include a glossary/list of abbreviations (SRP, GET, SND, EMC, NAC,...)
As suggested, we added a list of abbreviations at the end of the article.
2- In my pdf, text was deleted between l852 and l858
We apologize for this inconvenience. We will double-check during resubmission that all text lines and figure files will appear in the converted PDF.
3- l1183 - "Emcee"
According to the Merriam Webster dictionary “emcee“ is the noun that describes a person who acts as host for a program of entertainment (https://www.merriam-webster.com/dictionary/emcee). The word is synonymous to “master of ceremonies, MC, announcer, or host”. Considering the intended wordplay with regard to the EMC (ER membrane complex) that directs certain polypeptides into the ER membrane, we thought the term “emcee” as appropriate and decided to keep it.
4 - Not all references have a DOI link
Thank you for pointing this out. We added the missing DOIs.

Reviewer 2 Report
REPORT Tirincsi et al.
In recent years, the pathways that effect targeting of newly synthesized proteins from cytosolic ribosomes to the ER, and their subsequent insertion into- or translocation across the ER membrane, have turned out to be more diversified than was previously anticipated. Although a number of reviews have recently been written on this subject (cited in the submitted manuscript), this manuscript attempts to review all the current information on the different pathways relevant both to soluble lumenal and to membrane proteins, and is thus more comprehensive than other published reviews. As such, it could potentially serve as a useful unbiased reference paper for the cell/molecular biology community. Unfortunately, however, the manuscript has the serious drawback of being difficult and cumbersome to read, and some parts lack clarity. The English writing is generally quite good, however, there are some misuses of terms that need correction.
Below are some specific concerns and suggestions; however, I think that it would be useful for the authors to ask a colleague not directly involved in protein targeting to the ER and translocation to read the paper and make suggestions for improved clarity.
- While the title of the review states that it is focussed on the mammalian system, at various points comparisons with yeast are, understandably, made, as yeast studies have provided inestimable support for the progress in unravelling the mammalian pathways. For instance, ample space is given to comparison between the yeast and mammalian GET pathways (p. 14, 15, 26) and many yeast papers are cited. One wonders, then, why the authors have omitted the findings that weak signal peptides result in post-translational translocation in yeast. Specifically, on p. 2 it is stated that "two constraints stigmatize polypeptides to use post-translational targeting, a short overall precursor length (<100 amino acids) and C-terminal positioning of the targeting signal." Weak SPs in yeast should be mentioned here and at line 22, and reference(s) given (e.g., Plath and Rapoport, doi: 10.1083/jcb.151.1.167; Panzner et al., doi: 10.1016/0092-8674(95)90077-2). The posttranslational translocation of secretory proteins with weak signals is taken up on p. 8, but ignored on p. 2. In E. coli, secretion of proteins to the periplasmic place is prevalently post-translational, a finding that might also be mentioned, since the authors do discuss SecA in a subsequent section (p. 23).
Along the same lines, the yeast Snd pathway, the starting point for investigations on the role of mammalian SND2 in membrane protein targeting, is very poorly described. Since this pathway was discovered rather recently, it must be described in some more detail to allow readers to orient themselves, e.g., in section 4.3 dedicated to this pathway. It should also be stated here that the evidence for the role of this pathway is mainly genetic, and that it has not yet been characterized at the molecular level, as is the case for the other two pathways. Is anything known on the interaction of Snd1 with the nascent polypeptide? If not, this should be stated clearly. Does it have regions of homology with any other proteins? have structural predictions been made? Indeed, I notice that in section 4.4 there is no subsection dedicated to Snd1, implying that perhaps nothing is know about this. But this has to be stated somewhere. Once the reader has acquired a clearer idea of this pathway, the work on hSND2 could be better appreciated.
- Section 2, which describes how events within the ribosomal tunnel, and also outside of the RNC can affect targeting, is confusing, as targeting factors have not been discussed yet, making it extremely difficult to understand the text. This section should be placed after the discussion of targeting factors. Section 3 (on signal sequences) will become Section 2; section 4 (on targeting factors) will become section 3, and the current section 2 will become the last section of section 3, or a separate section 4, if the authors prefer. The title of this section ("Ribosomal protein synthesis") is quite uninformative; I suggest something like " Interactions of targeting factors with translating ribosomes before emergence of signal sequences", but the authors can probably think of something better than this.
In the current Section 2 (which I urge the authors to move), NAC is introduced without a word on what it is. Do the authors expect all readers to be familiar with NAC? Please define the acronym, and give a brief description of this factor. And describe better the problem of the specificity of SRP-mediated targeting, and how NAC improves the specificity. There are two models on how it does this: competition for a common binding site on the ribosome, or co-binding of the two factors, with NAC slowing signalless RNC -SRP complex binding to SR.
Also in this section the two sentences at lines 115 - 119 are unclear. "However, it is tempting to speculate that alternative ways of translation initiation influence the choice of a protein targeting pathway, so that both these aspects are intertwined"; the two aspects would be ribosome heterogeneity and alternative ways of translation initiation? This should be specified, because the reading is difficult and somewhat tortuous. What are these alternative ways of translation initiation? The conclusions of ref 36 should be better explained.
The sentence at lines 230-231 should also be modified (see point 7 below).
- Section 4.4.4. Cargo Capture by BAG6. Line 852 is truncated. The title of this subsection is a bit confusing, because it implies that BAG6, like the other described targeting factors of the section, is involved in the targeting process, rather than in ubiquitination and degradation of mislocalised membrane proteins.
- The main message of this review is that different pathways are required to handle the great variety of different clients. Actually, the flexibility of the different pathways is quite amazing, although it is true, as stressed by the authors, that no single pathway can handle ALL the clients. Nevertheless, as illustrated in Fig. 5, the Sec61 complex is involved in translocation/insertion of all secretory proteins and all membrane proteins, with the exception of Type III and Type IV proteins. Thus, it remains the central player and is flexible enough to engage other partners as needed, depending on the client, and this should be admitted in various parts of the text. For instance, the statement on lines 906-908 ("recent findings in the field of ER proteins import also nullify the concept of a universal ER protein translocase") is too strong, and might induce the reader to consider the Sec61 complex to be equally important to other players of the game.
- Section 5.1.2. The Growing Family of Sec61 Complex Associated Factors. Why has TRAM been forgotten?
- Section 4.4.2., lines 803-805. The meaning of this sentence is not clear to me. Also not clear is what is meant by the statement that ER TA proteins have negative charge at the C-terminus; sequence analyses of human ER TA proteins indicate that the charge of the C-terminal region is widely dispersed, from negative to positive (e.g., Borgese et al., doi: 10.1007/s10930-019-09845-4). The recent work of Fry et al. (doi: 10.1111/tra.12809) demonstrates that charge is important for discrimination between the ER and the outer mitochondrial membrane for those TA proteins that have low TMH hydrophobicity.
- Section 2.3. lines 230-234. The statement ("How clients with central targeting signals are recognized and delivered to the mammalian ER remains to be clarified" ) is not justified. There is no reason to postulate that SRP is hindered in its capacity to recognize internal hydrophobic stretches, as long as they are sufficiently distant from the C-terminus, as argued also in Ref. 23. Furthermore, work from the Weissman group (ref. 153) has demonstrated that in yeast (where the SND pathway was discovered) all membrane proteins, except for TA proteins, but including those with internal hydrophobic signals, are SRP clients under physiological conditions. Therefore, the SND pathway may represent a backup system in yeast, and maybe also in mammals too. But the statement on lines 230-231 is misleading. The subsequent sentence (lines 231 -234) is unclear for anyone who is not familiar with the paper by Leznicki and High (Ref. 62): the study reports that SGTA may assist SRP-mediated targeting by shielding downstream TMHs from aggregation, ubiquitination and degradation.
- Section 4.3, lines 682-683. The statement that Jan et al (ref. 175) observed co-translational transport independent of SRP is erroneous: the cited paper does not claim SRP-independent co-translational translocation: rather, it demonstrates that clients that are capable of translocating post-translationally do in fact utilize SRP when it is present.
- Figures:
Figure 5: The left column indicates targeting factors in grey lettering. Since the figure illustrates the mammalian systems, the presence of SND in this column is misleading, as the human homologue of yeast Snd1 has not been found so far. It should at least be distinguished from the other characterised targeting factors (e.g., the identified targeting factors could be in another colour, and SND kept in grey). In addition, in line 1 of column 1, the arrows emanating form SRP/SND point in opposite directions: the arrow from SND appears to point to a type II substrate. What does this mean? As far as I know, there is no evidence for involvement of SND in type II protein targeting, and plenty of evidence for the involvement of SRP! Also, in this figure, it should be somewhere indicated that CCCDC47 + Asterix constitutes the PAT complex.
Figure 3: Absissa of Panel A: "Amount of TMHs" should be replaced by "Number of TMHs".
- Section 6. Throughout this section, the authors use the terms "heterozygous" and "homozygous" to describe mutations. Mutations are neither heterozygous nor homozygous, rather they can be recessive or dominant. The terms heterozygous and homozygous refer to the genotype of individuals, indicating whether they carry one or two copies of the allele of interest. I presume that by heterozygous and homozygous the authors mean dominant and recessive. This must be corrected. The same error is at line 978 (section 5.1).
- Other unclear points:
Lines 84-86: the statement that "GET1/2 and EMC have a differentially shaped pore as is provided by the Sec61 complex" is confusing, as the Sec61 "pore" (or rather channel) opens on both sides of the bilayer, whereas the "pore" of GET1/2 and EMC does not. Better use distinct terms for these very distinct structures.
Line 200-201: ""the above-described mechanisms for nascent chain- and mRNA-dependent recruitment compensate for its low abundance and promote the occupation of most ribosomes with SRP". What is meant here is "the occupation by SRP of most ribosomes with SP-containing nascent peptides".
Lines 591-593: GET3 "transitions between open and closed conformations that allow cargo binding and release", the order of terms in "cargo binding and release" is inverted with respect to "open and closed conformations", so the reader might understand that cargo binds to GET3 in the open conformation and is released when GET3 is in the closed conformation.
Line 696: the term CAML is used without explanation. Throughout the text, the term GET2 has been used, and for sure the non-specialist reader will not understand what is being talked about here.
Line 1280: "Similar to SRP72, the loss-of-function of SRP54...." There is no description of loss of function mutations of SRP72 in the text. On line 1272 premature or missense mutations of SRP72 are described, but these mutations would seem to be dominant (heterozygous?), thus more likely (although not necessarily) gain-of-function.
Line 1289: "physiological phenotypes associated with BAG6, GET5 and SGTA". Maybe the authors mean "pathological"?
- References:
Line 506: for the "the vitality of SRP knock-out cells", the paper by Hahn and Walter (doi: 10.1016/0092-8674(91)90577-l) should be cited.
Line 1089: for post-translational N-glycosylation by the OST complex, Shrimal et al. (doi: 10.1083/jcb.201301031) should be cited.
Line 343: the transiently opening binding pocket of SRP is not described in Ref 12 (Walter et al., 1981) or Ref 81 (Walter and Blobel, 1980). Likewise, the paper of Deshaies and Schekman (Ref. 107) reports the first identification of Sec61p by a genetic screen. Structural studies would be better cited here (e.g., Keenan et al., doi: 10.1016/s0092-8674(00)81418-x ; van den Berg et al., doi: 10.1038/nature02218; Vorhees and Hegde, doi: 10.1126/science.aad4992).
Lines 484-485: Insertion of Cytb5 into protein-free membranes of liposomes was not shown in in Ref. 146. The correct reference is Brambillasca et al. (doi: 10.1038/sj.emboj.7600730).
Line 617: the reference on antagonism between BAG6 and SGTA is Wunderley et al., doi: 10.1242/jcs155648.
Line 947: for the structure of Sec61, van den Berg et al.,(doi: 10.1038/nature02218) and subsequent structural work should be cited.
- Misuse of some English terms and other:
First sentence: "Cells must be genius" - Genius must be used in the plural: "Cells must be geniuses"; the singular used in the Einstein quote refers to genius as a quality; in the first sentence of the ms it is used as predicative. Another possibility would be "Cells must have genius".
Line 56: "Two constraints stigmatize polypeptides to use post-translational targeting". The definition of "stigmatize" (from Word Reference) is: to put some mark of disgrace, shame, or dishonor upon (someone). Are proteins that undergo post-translational translocation/insertion dishonoured?
Line 93: "none size fits all"; "none is a pronoun and cannot be used as adjective of a noun; the correct sentence is "no one size fits all". This should be corrected throughout the text.
Line 242: Blobel's first name was Günter, not Günther.
Line 377: "What else targeting information"; presumably the authors mean "what other targeting information".
There are several other small grammatical errors, which the authors should check.
Author Response
Comments and Suggestions for Authors
REPORT Tirincsi et al.
In recent years, the pathways that effect targeting of newly synthesized proteins from cytosolic ribosomes to the ER, and their subsequent insertion into- or translocation across the ER membrane, have turned out to be more diversified than was previously anticipated. Although a number of reviews have recently been written on this subject (cited in the submitted manuscript), this manuscript attempts to review all the current information on the different pathways relevant both to soluble lumenal and to membrane proteins, and is thus more comprehensive than other published reviews. As such, it could potentially serve as a useful unbiased reference paper for the cell/molecular biology community. Unfortunately, however, the manuscript has the serious drawback of being difficult and cumbersome to read, and some parts lack clarity. The English writing is generally quite good, however, there are some misuses of terms that need correction.
We deeply appreciate the time and effort taken by reviewer 2 to critically assess the manuscript and providing such detailed as well as stimulating feedback. We tried to address most of the criticism raised and hope this will help to improve clarity of the manuscript.
Below are some specific concerns and suggestions; however, I think that it would be useful for the authors to ask a colleague not directly involved in protein targeting to the ER and translocation to read the paper and make suggestions for improved clarity.
- While the title of the review states that it is focussed on the mammalian system, at various points comparisons with yeast are, understandably, made, as yeast studies have provided inestimable support for the progress in unravelling the mammalian pathways. For instance, ample space is given to comparison between the yeast and mammalian GET pathways (p. 14, 15, 26) and many yeast papers are cited. One wonders, then, why the authors have omitted the findings that weak signal peptides result in post-translational translocation in yeast. Specifically, on p. 2 it is stated that "two constraints stigmatize polypeptides to use post-translational targeting, a short overall precursor length (<100 amino acids) and C-terminal positioning of the targeting signal." Weak SPs in yeast should be mentioned here and at line 22, and reference(s) given (e.g., Plath and Rapoport, doi: 10.1083/jcb.151.1.167; Panzner et al., doi: 10.1016/0092-8674(95)90077-2). The posttranslational translocation of secretory proteins with weak signals is taken up on p. 8, but ignored on p. 2. In E. coli, secretion of proteins to the periplasmic place is prevalently post-translational, a finding that might also be mentioned, since the authors do discuss SecA in a subsequent section (p. 23).
We agree with the points raised by the reviewer and have included the suggestions and corresponding references. Accordingly, we made the following changes:
- We rephrased the title and omitted the term “mammalian” to not provide the reader with the false assumption of solely focusing on mammalian systems.
- The set of key features that can determine a post-translational substrate was extended by the weak signal peptide of yeast substrates. We added the two references from the Rapoport lab. The passage reads:
“Key features of post-translationally targeted polypeptides can be a short overall precursor length (< 100 amino acids), a C-terminal positioning of a transmembrane helix (TMH) that serves as targeting signal, or, as is the case for yeast, a “weak” N-terminal signal peptide (SP) that is required for targeting [14-16].”
- The statement “Likewise, secretion of proteins to the periplasmic space of E. coli occurs prevalently post-translational.” was added to section 2.1.1.
Along the same lines, the yeast Snd pathway, the starting point for investigations on the role of mammalian SND2 in membrane protein targeting, is very poorly described. Since this pathway was discovered rather recently, it must be described in some more detail to allow readers to orient themselves, e.g., in section 4.3 dedicated to this pathway. It should also be stated here that the evidence for the role of this pathway is mainly genetic, and that it has not yet been characterized at the molecular level, as is the case for the other two pathways. Is anything known on the interaction of Snd1 with the nascent polypeptide? If not, this should be stated clearly. Does it have regions of homology with any other proteins? have structural predictions been made? Indeed, I notice that in section 4.4 there is no subsection dedicated to Snd1, implying that perhaps nothing is know about this. But this has to be stated somewhere. Once the reader has acquired a clearer idea of this pathway, the work on hSND2 could be better appreciated.
As suggested by the reviewer, we have provided a more detailed description of the pathway (section 3.3) addressing its identification, factors, and putative organization. A brief statement on putative cargo binding by the cytosolic component Snd1 is now made in the newly added section (section 3.4.5).
- Section 2, which describes how events within the ribosomal tunnel, and also outside of the RNC can affect targeting, is confusing, as targeting factors have not been discussed yet, making it extremely difficult to understand the text. This section should be placed after the discussion of targeting factors. Section 3 (on signal sequences) will become Section 2; section 4 (on targeting factors) will become section 3, and the current section 2 will become the last section of section 3, or a separate section 4, if the authors prefer. The title of this section ("Ribosomal protein synthesis") is quite uninformative; I suggest something like " Interactions of targeting factors with translating ribosomes before emergence of signal sequences", but the authors can probably think of something better than this.
Thank you for the suggestion. Accordingly, we rearranged and re-named the section(s). The new order starts with targeting signals (section 2) and is followed by targeting factors (section 3) and their interplay at the ribosomal exit tunnel (section 4).
In the current Section 2 (which I urge the authors to move), NAC is introduced without a word on what it is. Do the authors expect all readers to be familiar with NAC? Please define the acronym, and give a brief description of this factor. And describe better the problem of the specificity of SRP-mediated targeting, and how NAC improves the specificity. There are two models on how it does this: competition for a common binding site on the ribosome, or co-binding of the two factors, with NAC slowing signalless RNC -SRP complex binding to SR.
As stated above, we moved section 2 which is now placed after the introduction of targeting factors. Along with this change, the definition of the NAC acronym was complemented by a brief description of the factor. Also, more explicit explanation of its impact and the opposing roles of SRP and NAC was added. We hope that these changes provide better insights on NAC and improve the clarity of this section.
“Detailed understanding of which cytosolic factors interact with the ribosome and where on the ribosome is emerging and understood fairly well for SRP. It shares an overlapping binding site at the tunnel exit with NAC, which implies a competition of different factors for the egressing nascent chain. Unlike the far less abundant SRP, NAC actually exists in equimolar concentration to the ribosome and seems to scan nascent polypeptides by reaching into the exit tunnel [242,244]. Such a priority access enables NAC to orchestrate downstream factors. For example, the occupation of ribosomes by NAC prevents the unproductive, premature interaction between ribosomes synthesizing non-secretory proteins and the Sec61 complex [245]. On the other hand, SRP promotes the binding of ribosomes that synthesize secretory proteins with proper targeting signals to the Sec61 complex. In this regard, NAC is controversially discussed to modulate the binding affinity of SRP for non-secretory RNCs, thereby influencing the specificity of SRP-mediated ER targeting [217,246]. While on model excludes the simultaneous binding of NAC and SRP to the ribosome, a second model favors co-binding of both factors near the exit tunnel with NAC acting as negative regulator that selectively displaces parts of SRP from the exit port [247-249]. As a consequence, NAC hampers the efficient targeting of SRP to its receptor in the ER membrane in case of “signal sequence-less” RNCs and other factors may gain access to the cargo. Besides the regulation by NAC, the mechanisms for nascent chain- and mRNA-dependent recruitment (Figure 4) compensate for the low abundance of SRP and prioritize the occupation of secretory RNCs with SRP [191,222,234].”
Also in this section the two sentences at lines 115 - 119 are unclear. "However, it is tempting to speculate that alternative ways of translation initiation influence the choice of a protein targeting pathway, so that both these aspects are intertwined"; the two aspects would be ribosome heterogeneity and alternative ways of translation initiation? This should be specified, because the reading is difficult and somewhat tortuous. What are these alternative ways of translation initiation? The conclusions of ref 36 should be better explained.
We thank the reviewer for bringing up this matter, giving us the chance to clarify the text on this subject. The two aspects the sentence is referring to are alternative ways of translation initiation on the one hand and alternative ways of protein targeting on the other hand. The sentence speculates about a putative correlation and functional connection between specific modes of translation initiation and protein targeting, so that the individual mechanism of translation initiation may affect how subsequent targeting of the protein is mediated. Diverse mechanisms of alternative translation initiation were recently reviewed by James and Smyth (James and Smyth 2018; doi: 10.1016/j.lfs.2018.09.054). Some of these mechanisms involve elements in the mRNA, such as internal ribosome entry sites (IRES) or RNA hypoxia response elements (rHREs), which may prevalently occur in cargos that utilize the same pathway for targeting to the ER, e.g., an SRP-independent route (Uniacke et al. 2012; doi: 10.1038/nature11055; Melanson et al. 2017; doi: 10.1155/2017/6098107). Accordingly, bringing the corresponding targeting factors under the same control element would allow coordinated regulation of both processes, targeting and translation. Likewise, distinct elements in the mRNA can dictate the choice of a targeting pathway by recruitment of a specific targeting factor, as has been described for certain 3’ untranslated regions and SRP (Chartron et al. 2016; doi:10.1038/nature19309). Specialized ribosomes may thus be primed for a specific targeting pathway and selective translation of specific mRNAs coding for respective clients.
In line with the above explanations and refining the conclusion of the initial ref.36 (Chartron et al. 2016; doi:10.1038/nature19309), we added missing information to the text and hope that it will make this note clearer:
“These involve elements in the mRNA, such as internal ribosome entry sites or RNA hypoxia response elements (rHREs), which may prevalently occur in cargos that utilize the same ER targeting route [219-221]. Endowing the corresponding targeting factors with the same mRNA control element (e.g., rHRE) would allow the coordinated regulation of both processes, translation and targeting (cf., oxygen-regulated transcription of HSND2 [211]). Likewise, distinct elements in the mRNA may directly dictate the choice of a targeting pathway by recruitment of a certain targeting factor. Such capacity has indeed been described for mRNA with certain 3’ untranslated regions and SRP [222]. Specialized ribosomes may thus be primed for a specific targeting pathway and selective translation of specific mRNAs coding for respective clients.”.
The sentence at lines 230-231 should also be modified (see point 7 below).
Taking the reviewer`s advice, we modified the sentence as follows “Questions regarding which factor catches the cargo first and why an additional component, BAG6, is located at the mammalian ribosome are currently unclear.”
- Section 4.4.4. Cargo Capture by BAG6. Line 852 is truncated. The title of this subsection is a bit confusing, because it implies that BAG6, like the other described targeting factors of the section, is involved in the targeting process, rather than in ubiquitination and degradation of mislocalised membrane proteins.
Although the capture of cargo by BAG6 may not be associated with targeting, rather than degradation, we find that its competitive binding to GET clients might still contribute to their targeting fate. As such, we prefer to keep the title of this subsection for reasons of consistency and instead, added further explanation to the text in section 3.4.4:
“Although capture of cargo by BAG6 seems not to be directly associated with targeting, rather than degradation, its competitive binding to GET clients might still contribute to their targeting fate”.
- The main message of this review is that different pathways are required to handle the great variety of different clients. Actually, the flexibility of the different pathways is quite amazing, although it is true, as stressed by the authors, that no single pathway can handle ALL the clients. Nevertheless, as illustrated in Fig. 5, the Sec61 complex is involved in translocation/insertion of all secretory proteins and all membrane proteins, with the exception of Type III and Type IV proteins. Thus, it remains the central player and is flexible enough to engage other partners as needed, depending on the client, and this should be admitted in various parts of the text. For instance, the statement on lines 906-908 ("recent findings in the field of ER proteins import also nullify the concept of a universal ER protein translocase") is too strong, and might induce the reader to consider the Sec61 complex to be equally important to other players of the game.
We have toned down the opening statement in section 5 to: “Consistent with the multiplicity and complexity of targeting signals and targeting pathways described in the previous sections, recent findings in the field of ER protein import also extend this concept to a small assortment of ER protein translocases. Different types and arrangements of protein translocases, sometimes acting in concert, manage the translocation or insertion of a dedicated subset of incoming polypeptides (Figure 5).”
In addition, we have interspersed and adjusted a few statements that highlight the Sec61 complex as the major protein translocation device of the ER membrane. For example, the argument section 5.1 states:
”Subsequently, the mammalian homolog of Sec61p was identified and its functional as well as structural characterization as the major polypeptide-conducting channel of the ER followed”.
Similarly, the headline for paragraph 5.1. includes the hint “The Sec61 Translocase - Director for SPs and the Majority of TMHs”.
- Section 5.1.2. The Growing Family of Sec61 Complex Associated Factors. Why has TRAM been forgotten?
We have added TRAM1 to the paragraph about the Sec61 auxiliary factors. “Using classical reconstitution approaches, the translocating chain-associated membrane protein 1 (TRAM1) has been found to aid the insertion of nascent chains into the Sec61 complex when their SP has a shorter than average N-region [318,335,337,338]. Further data imply that TRAM1 may also assist protein import by making the lipid bilayer in vicinity of the lateral gate of the Sec61 complex conducive for accepting targeting signals [339].”.
- Section 4.4.2., lines 803-805. The meaning of this sentence is not clear to me. Also not clear is what is meant by the statement that ER TA proteins have negative charge at the C-terminus; sequence analyses of human ER TA proteins indicate that the charge of the C-terminal region is widely dispersed, from negative to positive (e.g., Borgese et al., doi: 10.1007/s10930-019-09845-4). The recent work of Fry et al. (doi: 10.1111/tra.12809) demonstrates that charge is important for discrimination between the ER and the outer mitochondrial membrane for those TA proteins that have low TMH hydrophobicity.
As suggested, the sentence of the previous section 4.4.2 was rephrased for clarification and now reads:
“Also, helix 8 does not play a prominent role in this model. However, it solves the mechanistic dilemma of an exclusively closed Get3-cargo complex while allowing high-affinity binding of cargo with dynamic and directed hand-over between factors”.
With respect to statement previously found in line 828, we absolutely agree with the reviewer’s point of view, as we have also stated elsewhere in the manuscript. While the charge of the C-terminal region is widely dispersed considering all TA proteins, in the individual case of cytochrome b5, however, a negatively charged C-terminus increased specificity of targeting to the ER (Borgese et al. 2001; doi: 10.1091/mbc.12.8.2482). In the interest of simplification, we shortened the last section of the paragraph, which now reads:
“Nevertheless, the physicochemical properties of TA-TMHs from ER, mitochondria, and peroxisomes are overlapping [153], yet the capacity to detect and select for differential C-terminal charges seems to be missing in the GET pathway [156]. Yeast Get3 was indeed found to bind and deliver mitochondrial TA proteins to the ER, which led to the conclusion that an as of yet undefined mitochondrial targeting pathway might outcompete the GET pathway under cellular conditions [204].”.
- Section 2.3. lines 230-234. The statement ("How clients with central targeting signals are recognized and delivered to the mammalian ER remains to be clarified" ) is not justified. There is no reason to postulate that SRP is hindered in its capacity to recognize internal hydrophobic stretches, as long as they are sufficiently distant from the C-terminus, as argued also in Ref. 23. Furthermore, work from the Weissman group (ref. 153) has demonstrated that in yeast (where the SND pathway was discovered) all membrane proteins, except for TA proteins, but including those with internal hydrophobic signals, are SRP clients under physiological conditions. Therefore, the SND pathway may represent a backup system in yeast, and maybe also in mammals too. But the statement on lines 230-231 is misleading. The subsequent sentence (lines 231 -234) is unclear for anyone who is not familiar with the paper by Leznicki and High (Ref. 62): the study reports that SGTA may assist SRP-mediated targeting by shielding downstream TMHs from aggregation, ubiquitination and degradation.
We agree with the notion that SRP can bind internal hydrophobic stretches and revised the text to acknowledge the respective findings. Also, the reference to the study of Leznicki and High was rephrased. The passage in question now reads:
“How the corresponding clients with a central TMH are recognized and delivered to the mammalian ER thus remains to be clarified. In principle, several factors have the capacity to bind central targeting signals. First, SRP was described to be involved in all membrane protein targeting in yeast, including those with TMHs distant from the termini [125]. Second, SGTA may also bind internal TMHs, though it was proposed to assist SRP-mediated targeting by preventing aggregation, ubiquitination, and degradation [150].”.
- Section 4.3, lines 682-683. The statement that Jan et al (ref. 175) observed co-translational transport independent of SRP is erroneous: the cited paper does not claim SRP-independent co-translational translocation: rather, it demonstrates that clients that are capable of translocating post-translationally do in fact utilize SRP when it is present.
We apologize for this misinterpretation and have rephrased the argument and references to provide a better suited explanation. The sentence now reads:
“Congruent with the observation of protein transport in the absence of SRP and GET [120,121], a genetic screen in yeast uncovered the SND (SRP independent) pathway as al-ternative targeting route capable of delivering a broad range of substrates to the ER.”
[Ref. 120 & 121: Ast and Schuldiner 2013, doi:10.3109/10409238.2013.782999; Ast et al. 2013, doi:10.1016/j.cell.2013.02.003.]
- Figures:
Figure 5: The left column indicates targeting factors in grey lettering. Since the figure illustrates the mammalian systems, the presence of SND in this column is misleading, as the human homologue of yeast Snd1 has not been found so far. It should at least be distinguished from the other characterised targeting factors (e.g., the identified targeting factors could be in another colour, and SND kept in grey). In addition, in line 1 of column 1, the arrows emanating form SRP/SND point in opposite directions: the arrow from SND appears to point to a type II substrate. What does this mean? As far as I know, there is no evidence for involvement of SND in type II protein targeting, and plenty of evidence for the involvement of SRP! Also, in this figure, it should be somewhere indicated that CCCDC47 + Asterix constitutes the PAT complex.
Thank you for pointing this out. We have made the changes and updated both the figure as well as the legend to clarify the major points raised by the reviewer. The new legends reads:
Figure 5. Features of protein translocation machines and their substrate spectrum. Depicted from the left to right are targeting pathways (blue), membrane-integrated protein translocation machines (pink, purple), their favored types of clients (boxes), and some key features for the central components of the protein translocation machines. With the exception of GET1/2, the membrane-anchored receptor components for the targeting pathways are not shown. The scarcely characterized SND pathway (grey) as alternative targeting route for a subset of GPI-anchored, tail-anchored, short secretory, and polytopic membrane proteins is indicated ones. If SND delivers substrates either to a preferred ER translocase or different ones, or if its membrane-embedded component(s) might perform a dual function as receptor plus insertase similar to the GET1/2 complex, remains to be seen. The auxiliary TMCO1 and PAT complexes are presented as one assembly. Of note, the PAT complex that was also described as stand-alone intramembrane chaperone complex is a heterodimer comprised of the indicated proteins CCDC47 and Asterix [251]. Signal peptides and transmembrane helices are integrated into the ER membrane and classified and colored according to Figures 1 and 2. The cleavage of signal peptides for secretory and type I membrane proteins is indicated (scissors and cleaved signal peptides are shown) as is the C-terminus (C) for each type of membrane protein. The spectrum of substrates preferentially handled by assemblies that entail the Sec61 complexes are represented by grey boxes. Substrates handled by the ER membrane complex (EMC) and the GET1/2 complex are highlighted by light blue and yellow boxes, respectively. Overlapping boxes depict overlaps in the substrate range.
Figure 3: Absissa of Panel A: "Amount of TMHs" should be replaced by "Number of TMHs".
Done.
- Section 6. Throughout this section, the authors use the terms "heterozygous" and "homozygous" to describe mutations. Mutations are neither heterozygous nor homozygous, rather they can be recessive or dominant. The terms heterozygous and homozygous refer to the genotype of individuals, indicating whether they carry one or two copies of the allele of interest. I presume that by heterozygous and homozygous the authors mean dominant and recessive. This must be corrected. The same error is at line 978 (section 5.1).
Pardon this mistake of incorrect nomenclature and sloppiness on our end. The reviewer is correct and we have updated the paragraph. Upon re-inspection of various literature references the error in describing mutations as heterozygous/homozygous seems to be more prevalent than anticipated. Thanks for bringing this to our attention so we can try to avoid that mistake in the future.
- Other unclear points:
Lines 84-86: the statement that "GET1/2 and EMC have a differentially shaped pore as is provided by the Sec61 complex" is confusing, as the Sec61 "pore" (or rather channel) opens on both sides of the bilayer, whereas the "pore" of GET1/2 and EMC does not. Better use distinct terms for these very distinct structures.
We have re-phrased the sentence to avoid any confusion. The new version is “Interestingly, precursor proteins relying on the GET1/2 complex or EMC require a differentially shaped opening compared to the membrane-spanning pore that is provided by the Sec61 complex. This shows that nature found more than one solution for polypeptides to traverse a membrane.” In the later passages and figures we try to explain the difference in design of the “pore” more precisely by terms like “discontinuous vestibule”, “hydrophilic groove”, or “hemi-channel”.
Line 200-201: ""the above-described mechanisms for nascent chain- and mRNA-dependent recruitment compensate for its low abundance and promote the occupation of most ribosomes with SRP". What is meant here is "the occupation by SRP of most ribosomes with SP-containing nascent peptides".
As we would like to keep the option of mRNA-dependent recruitment of SRP to ribosomes as part of the statement, we have adjusted it as follows:
“the mechanisms for nascent chain- and mRNA-dependent recruitment (Figure 4) compensate for the low abundance of SRP and prioritize the occupation of secretory RNCs with SRP [191,222,234].”
Lines 591-593: GET3 "transitions between open and closed conformations that allow cargo binding and release", the order of terms in "cargo binding and release" is inverted with respect to "open and closed conformations", so the reader might understand that cargo binds to GET3 in the open conformation and is released when GET3 is in the closed conformation.
The order of terms has been changed accordingly (closed and open – binding and release).
Line 696: the term CAML is used without explanation. Throughout the text, the term GET2 has been used, and for sure the non-specialist reader will not understand what is being talked about here.
We aimed to adhere to the recently revised nomenclature for the GET components and have overlooked this instance. We changed CAML to GET2. Thank you for picking this up.
Line 1280: "Similar to SRP72, the loss-of-function of SRP54...." There is no description of loss of function mutations of SRP72 in the text. On line 1272 premature or missense mutations of SRP72 are described, but these mutations would seem to be dominant (heterozygous?), thus more likely (although not necessarily) gain-of-function.
We rephrased the statement to: “Autosomal dominant mutations in SRP54 are known from three different patients and result in neutropenia with similarities to the Shwachman-diamond-syndrome. The three de novo missense variants of SRP54 all affect conserved residues of the GTPase domain known to be critical for GTP and receptor binding. In two of the three patients their neutropenia, due to the SRP54 mutation, was also accompanied by an exocrine pancreatic insufficiency.”
Line 1289: "physiological phenotypes associated with BAG6, GET5 and SGTA". Maybe the authors mean "pathological"?
We rephrased the sentences to: “Regarding the subunits of the GET pre-targeting complex BAG6 and co-chaperone SGTA, we like to refer readers to a recent review on the roles of these cytosolic quality control proteins in disease [393].”
- References:
Line 506: for the "the vitality of SRP knock-out cells", the paper by Hahn and Walter (doi: 10.1016/0092-8674(91)90577-l) should be cited.
We added this valuable resource.
Line 1089: for post-translational N-glycosylation by the OST complex, Shrimal et al. (doi: 10.1083/jcb.201301031) should be cited.
We added this valuable resource.
Line 343: the transiently opening binding pocket of SRP is not described in Ref 12 (Walter et al., 1981) or Ref 81 (Walter and Blobel, 1980). Likewise, the paper of Deshaies and Schekman (Ref. 107) reports the first identification of Sec61p by a genetic screen. Structural studies would be better cited here (e.g., Keenan et al., doi: 10.1016/s0092-8674(00)81418-x ; van den Berg et al., doi: 10.1038/nature02218; Vorhees and Hegde, doi: 10.1126/science.aad4992).
We exchanged the references accordingly. It was not the intention to cite structural studies of the binding pockets rather than original papers about the “first elements” of protein translocation (SRP and Sec61) in more general terms.
Lines 484-485: Insertion of Cytb5 into protein-free membranes of liposomes was not shown in in Ref. 146. The correct reference is Brambillasca et al. (doi: 10.1038/sj.emboj.7600730).
As different C-terminal extensions of Cytb5 were tested in presence of liposomes in both studies, we added the one mentioned by the reviewer, but also kept the reference in use.
Line 617: the reference on antagonism between BAG6 and SGTA is Wunderley et al., doi: 10.1242/jcs155648.
We added the reference as suggested.
Line 947: for the structure of Sec61, van den Berg et al.,(doi: 10.1038/nature02218) and subsequent structural work should be cited.
The numbering of lines in our document seem not to align perfectly with the ones from the reviewer. We have referenced many structural studies on the Sec61 complex. One text passage around line 947 reads: “Subsequently, the mammalian homolog of Sec61p was identified and its functional as well as structural characterization as the major polypeptide-conducting channel of the ER followed [79,257-260]. Complementing the functional conservation, many structural studies also highlight the conserved architecture of the Sec61 complex from bacteria and archaea to lower and higher eucaryotes [261-267].” The references with regard to structural studies include van den Berg et al. 2004 [#79] and subsequent studies by further authors and labs. By no means have we had the intention to deliberately exclude relevant publications.
- Misuse of some English terms and other:
First sentence: "Cells must be genius" - Genius must be used in the plural: "Cells must be geniuses"; the singular used in the Einstein quote refers to genius as a quality; in the first sentence of the ms it is used as predicative. Another possibility would be "Cells must have genius".
We deleted the opening statement and quotes and went with a more conservative version that now reads: “Eucaryotic cells use the principle of compartmentalization to streamline the flow of information within the crowded intracellular environment.”.
Line 56: "Two constraints stigmatize polypeptides to use post-translational targeting". The definition of "stigmatize" (from Word Reference) is: to put some mark of disgrace, shame, or dishonor upon (someone). Are proteins that undergo post-translational translocation/insertion dishonoured?
In accordance with the previous comment #1 the sentence and word choice was changed to “Key features of post-translationally targeted polypeptides can be a short overall precursor length (< 100 amino acids), a C-terminal positioning of a transmembrane helix (TMH) that serves as targeting signal, or, as is the case for yeast, a “weak” N-terminal signal peptide (SP) that is required for targeting”.
Line 93: "none size fits all"; "none is a pronoun and cannot be used as adjective of a noun; the correct sentence is "no one size fits all". This should be corrected throughout the text.
Thank you for pointing this out. We introduced the corresponding change and also deleted the statement in one instance.
Line 242: Blobel's first name was Günter, not Günther.
We regret this mistake and have corrected the misspelling.
Line 377: "What else targeting information"; presumably the authors mean "what other targeting information".
We agree, the wording of the sentence was off. For reasons of simplicity, we deleted the sentence plus the following two.
There are several other small grammatical errors, which the authors should check.
We tried to correct such errors to the best of our ability.

Round 2
Reviewer 2 Report
The authors have done a very thorough job in addressing my many concerns on the previous version, and in doing so have substantially improved the manuscript, which is now suitable for publication, pending correction of a few typos/misspelling and additional unclear points that I noticed, as follows:
line 584: with the term "later", the authors presumably mean "latter" (which means the last mentioned - "later" instead has a temporal meaning, being the comparative of "late")
lines 594- 595: GPI proteins are not of low to moderate hydrophobicity: it is their initial C-terminal segment, which is then removed, that has characteristically low-moderate hydrophobicity. The addition of "precursors" (GPI protein precursors) should fix this confusion.
lines 675-677: " In animals, helix 8 counts three methionine residues along ~21 amino acids, while there are only two methionine 15his15ues in the15 amino acids of S. cerevisiae"; replace "along" with "among"; what does 15his15ues mean? "in the15 amino acids of S. cerevisiae" replace with "among the15 amino acids of the S. cerevisiae homologous helix".
line 698: What is the 16his model?
line 965: While one model (current text is "while on model")
line 1044: ".. and polytopic proteins is indicated ones"; I suppose that the authors mean "once" rather than "ones"
Figure 5: This reviewer still find the description of the PAT complex/TMCO1 complex confusing for the reader that is not well versed in the field. In the grey box of column 3 TMCO1 and the PAT complex are indicated, however in the subsequent column it is not clear which proteins comprise the PAT complex, and what complex Nicalin, TMEM147 and NOMO belong to. The three lines of this description should be organized differently, i.e.:
first line: TMCO1
second line: TMEM 147, NOMO, Nicalin (TMEM147 complex)
third line: PAT complex (CCDC47, Asterix).
Finally, as I remember, the other reviewer of this manuscript suggested providing a table with a list of the many acronyms used; I think that this is a very good idea, and urge the authors to add such a table to the paper.
Author Response
The authors have done a very thorough job in addressing my many concerns on the previous version, and in doing so have substantially improved the manuscript, which is now suitable for publication, pending correction of a few typos/misspelling and additional unclear points that I noticed, as follows:
=> We are grateful that reviewer 2 had another close look at the edited version of the manuscript and points out some helpful corrections. Thank you. We addressed the points raised and deleted some mysterious additions like “15his15ues”, which should state “residues”. Pardon this flaw.
line 584: with the term "later", the authors presumably mean "latter" (which means the last mentioned - "later" instead has a temporal meaning, being the comparative of "late")
=> In this case, the word “later” was intended to be used with its temporal connotation. The original screen (Pan et al. 2006, doi:10.1016/j.cell.2005.12.036) did not refer to the genes by its SND nomenclature which was established only later. We decided to exclude the term “later” from the sentence to make it more reader-friendly:
“Initial evidence for the involvement of the SND2 and SND3 genes in protein targeting arose from another yeast screen showing synthetic lethality when deletions of either component were combined with deletions of the GET2 or GET3 gene [184].”
lines 594- 595: GPI proteins are not of low to moderate hydrophobicity: it is their initial C-terminal segment, which is then removed, that has characteristically low-moderate hydrophobicity. The addition of "precursors" (GPI protein precursors) should fix this confusion.
=> The reviewer is correct; it is the C-terminal GPI-attachment signal that is of low hydrophobicity rather than the entire GPI-anchored protein. We made the add on to improve clarity of the statement, which now reads:
”While a detailed mechanistic description of the pathway is still missing, the currently known client spectrum in mammalian cells comprises co-translational cargo such as membrane proteins with internal TMHs and GPI-anchored precursor proteins with a GPI attachment signal of low hydrophobicity [22,43,185].
lines 675-677: " In animals, helix 8 counts three methionine residues along ~21 amino acids, while there are only two methionine 15his15ues in the15 amino acids of S. cerevisiae"; replace "along" with "among"; what does 15his15ues mean? "in the15 amino acids of S. cerevisiae" replace with "among the 15 amino acids of the S. cerevisiae homologous helix".
=> We made the corrections as suggested. 15his15ues was replaced by residues. The sentence now reads:
“In animals, helix 8 counts three methionine residues among 21 amino acids, while there are only two methionine residues among the 15 amino acids of the homologous helix in S. cerevisiae [195].”
line 698: What is the 16his model?
=> Changed to “this model”.
line 965: While one model (current text is "while on model")
=> Thanks, we added the missing “e”.
line 1044: ".. and polytopic proteins is indicated ones"; I suppose that the authors mean "once" rather than "ones"
=> The reviewer is correct, it should state “once”. We corrected the mistake.
Figure 5: This reviewer still find the description of the PAT complex/TMCO1 complex confusing for the reader that is not well versed in the field. In the grey box of column 3 TMCO1 and the PAT complex are indicated, however in the subsequent column it is not clear which proteins comprise the PAT complex, and what complex Nicalin, TMEM147 and NOMO belong to. The three lines of this description should be organized differently, i.e.:
first line: TMCO1
second line: TMEM 147, NOMO, Nicalin (TMEM147 complex)
third line: PAT complex (CCDC47, Asterix).
=> We regret that figure 5 still caused trouble and was difficult to comprehend. We took a slightly different approach than suggested by the reviewer. First, we extended the legend by the following sentences:
“The TMCO1 containing translocon comprises the Sec61 channel and five accessory factors: TMCO1, CCDC47, and the Nicalin-TMEM147-NOMO complex [252]. For reasons of simplicity, we refer to those five accessory factors as “TMCO1 complex”. The PAT and TMCO1 complex share at least one subunit, CCDC47. Further details of the TMCO1-PAT-Sec61 assembly and how they might act in concert to form an operational protein translocase can be found in the text.”
Second, we made use of the space in the lower right corner of the figure and devoted a couple of lines to the subunit composition of the PAT and (what we refer to as) TMCO1 complex.
We hope the reviewer finds these additions suitable to portray the subunit composition of the PAT and TMCO1 complexes.
Finally, as I remember, the other reviewer of this manuscript suggested providing a table with a list of the many acronyms used; I think that this is a very good idea, and urge the authors to add such a table to the paper.
=> As suggested before, we did add a list of abbreviations. Currently this list can be found at the end of the article between the conflict-of-interest statement and the references. We will check with the production team or editor, where this list should be placed to adhere to the manuscript layout of the IJMS.
Thank you again for providing useful comments for improvement!
